# In-Context Learning as Conditioned Associative Memory Retrieval

**Weimin Wu** [* 1 2]   **Teng-Yun Hsiao** [* 3]   **Jerry Yao-Chieh Hu** [* 1 2]   **Wenxin Zhang** [2]   **Han Liu** [1 2 4]

## Abstract

We provide an exactly solvable example for interpreting In-Context Learning (ICL) with one-layer attention models as conditional retrieval of dense associative memory models. Our main contribution is to interpret ICL as memory reshaping in the modern Hopfield model from a conditional memory set (in-context examples). Specifically, we show that the in-context sequential examples induce an effective reshaping of the energy landscape of a Hopfield model. We integrate this *in-context memory reshaping* phenomenon into the existing Bayesian model averaging view of ICL [Zhang et al., AISTATS 2025] via the established equivalence between the modern Hopfield model and transformer attention. Under this unique perspective, we not only characterize how in-context examples shape predictions in the Gaussian linear regression case, but also recover the known $\epsilon$-stability generalization bound of the ICL for the one-layer attention model. We also give explanations for three key behaviors of ICL and validate them through experiments.

## 1 Introduction

We show that In-Context Learning (ICL) reshapes a model's internal associative memory using prompt examples, analogous to how a modern Hopfield network adjusts its stored memories in response to new cues. To be precise, we provide an exactly solvable example for interpreting ICL in one-layer attention models as conditional retrieval of dense

associative memory models (Krotov, 2023). With the success of large language models (Guo et al., 2025; Grattafiori et al., 2024; Jiang et al., 2024a), ICL enables large language models to adapt to diverse tasks by leveraging task-specific examples embedded in the input prompts, without updating model parameters (Min et al., 2022; Garg et al., 2022; Brown et al., 2020). This "learning in the forward pass" raises fundamental questions about how models represent and reuse knowledge, and challenges our understanding of how a fixed model can implement new task behaviors on the fly (Von Oswald et al., 2023; Akyürek et al., 2022; Xie et al., 2022). Understanding that mechanism is theoretically compelling and practically valuable: better mental models of ICL can guide prompt construction and improve the reliability of LLM-based systems or Transformer models in general (Zhang et al., 2025; Wu et al., 2025; Han et al., 2024; Bai et al., 2023; Li et al., 2023; Xie et al., 2022).

In this work, we approach ICL through the lens of associative memory (Krotov, 2023; Krotov & Hopfield, 2020; 2016; Hopfield, 1982; Amari, 1972). In particular, we draw inspiration from *transformer-compatible* modern Hopfield networks (Ramsauer et al., 2020). These networks are continuous-state dense associative memories with an energy landscape that stores memory patterns as local minima. Recent studies establish a formal equivalence between the transformer's attention mechanism and the update dynamics of a modern Hopfield network (Santos et al., 2024a;b; Hu et al., 2024b; Wu et al., 2024b; Krotov & Hopfield, 2020; Ramsauer et al., 2020). In a Hopfield network, a query pattern is iteratively matched to the closest stored memory pattern (a local minimum of the energy function). Analogously, a transformer's attention module can be viewed as retrieving a stored vector (from the key–value pairs) that best matches a query embedding. Consequently, training attention weights can be viewed as shaping the Hopfield energy landscape so that training examples become stable memories (Wu et al., 2024a; Hu et al., 2024a). This perspective suggests that a transformer may be remembering and retrieving relevant information from its weights when performing ICL, akin to an associative memory recalling patterns in response to cues. This unique perspective recasts prompt-based inference as a memory retrieval process, potentially offering a more concrete mechanism to explain how in-context examples influence model predictions.

---

[*]Equal contribution  [1]Center for Foundation Models and Generative AI, Northwestern University, USA [2]Department of Computer Science, Northwestern University, USA [3]Department of Physics, National Taiwan University, Taiwan [4]Department of Statistics and Data Science, Northwestern Universit, USA. Correspondence to: Weimin Wu <wwm@u.northwestern.edu>, Teng-Yun Hsiao <b10502058@ntu.edu.tw>, Jerry Yao-Cheih Hu <jhu@u.northwestern.edu>, Wenxin Zhang <wenxinzhang2025@u.northwestern.edu>, Han Liu <hanliu@northwestern.edu>.

*Proceedings of the $42^{nd}$ International Conference on Machine Learning*, Vancouver, Canada. PMLR 267, 2025. Copyright 2025 by the author(s).

Building on this connection, we posit to interpret ICL as a conditioned memory retrieval process in a dense associative memory model (in particular, a modern Hopfield model). Surprisingly, when each in-context example acts as a query to the Hopfield model, we show that the sequence of examples induces an *effective reshaping* of the energy landscape: attractors relevant to the prompt move closer to the query, while irrelevant ones recede. Although the permanent weights are unchanged, the effective memory is biased toward the task at hand, so the final test query retrieves information consistent with the demonstrated patterns. This in-context memory reshaping explains why in-context learning improves prediction without fine-tuning.

Concretely, we analyze an exactly solvable one-layer attention model. The model reveals that memory reshaping corresponds to a linear transformation of both key and value matrices determined by the prompt, and it links this mechanism to the Bayesian model averaging view of in-context learning (Zhang et al., 2025). Thus, we bridge an abstract Bayesian statistical explanation with analytic network dynamics from neuroscience. Under this unique perspective, we derive new insights and recover key results about ICL.

**Contributions.** Our contributions are as follows:

- **In-Context Memory Reshaping.** We present an exact formulation of *in-context memory reshaping* for ICL using a one-layer attention model. Under this stylized setting, we prove that an input prompt of $t$ in-context examples induces an *implicit* linear mapping on the model's pre-trained attention weights (comparing scenarios *without* and *with* prompt examples). Effectively, this reshaping modulates the model's stored knowledge without any parameter updates. Together, these insights offer a model-based explanation for ICL's internal mechanism and its effects in a simple attention.

- **Link to Bayesian Theory.** We embed this *in-context memory reshaping* phenomenon into the Bayesian model averaging framework for ICL (Zhang et al., 2025). This connection yields a solvable example that complements existing Bayesian theory of ICL and highlights how memory reshaping provides a complementary, mechanistic interpretation of the same behavior.

- **New Insights and Validations.** We derive and validate new insights: (i) In Gaussian linear regression, prompts steer predictions toward the Bayes-optimal solution by dynamically re-weighting stored memories. This serves as an explicit characterization of how prompts "guide" predictions. (ii) We recover the known $\epsilon$-stability bound for ICL (Li et al., 2023) within one-layer attention. This confirms that models following our memory retrieval interpretation achieve the same established generalization guarantees for ICL.

(iii) We give explanations for three key behaviors of ICL and validate through synthetic and real datasets. These behaviors include sensitivity to covariance shifts, sensitivity to response's accuracy, and sensitivity to the similarity between prompts and *test query*.

In sum, we interpret ICL as conditioned associative-memory retrieval. The resulting theory and experiments provide a unified and concrete account of how prompting examples modulate a transformer's computations on the fly.

**Organization.** Section 2 introduces the background on large language models, in-context learning, memory reshaping in modern Hopfield models, and related work. Section 3 shows the details of our proposed in-context memory reshaping phenomenon. Section 4 provides experimental studies. Section 5 provides discussion and conclusion.

**Notations.** We denote the set $\{1, ..., N\}$ as $[N]$. For a Polish space $\mathcal{S}$, let $\Delta(\mathcal{S})$ denote the set of all probability measures over $\mathcal{S}$. For a matrix $M \in \mathbb{R}^{d_1 \times d_2}$ ($d_1, d_2 \in \mathbb{N}^+$), we define the $l_{p,q}$ norm as $\|M\|_{p,q} = (\sum_{i=1}^{d_2} \|M_i\|_p^q)^{1/q}$, where $M_i \in \mathbb{R}^{d_1}$ denotes the $i$-th column of $M$. Let $c, r$ denote the query and response. For simplicity, we consider the setting where both $c$ and $r$ are single tokens.

## 2 Background and Related Work

In this section, we provide background on the following topics: (i) the formulation of ICL in large language models in Section 2.1, (ii) existing theory for ICL based on generic latent variable models in Section 2.2, and (iii) memory reshaping in modern Hopfield models in Section 2.3. We also review related work in Section 2.4.

### 2.1 In-Context Learning in Large Language Model

Let $P_\theta$ be a pre-trained large language model with parameters $\theta$. We consider the scenario where both queries and responses are single tokens in the large language model. Let $c \in \mathbb{R}^{d_c}$ and $r \in \mathbb{R}^{d_r}$ denote the query and response. Given $T \in \mathbb{N}^+$, for any $t \in [T]$, we denote the sequence of $t$ prompt examples (query-response pairs) as $\mathcal{D}_t := \{c_i, r_i\}_{i=1}^t$. Let $C_t = [c_1, \ldots, c_t] \in \mathbb{R}^{d_c \times t}$ and $R_t = [r_1, \ldots, r_t] \in \mathbb{R}^{d_r \times t}$. In ICL, the input to the pre-trained large language model is the prompt $P_t = (\mathcal{D}_t, c_{t+1})$, where $c_{t+1}$ is the $(t + 1)$-th query, also known as the *test* query. The large language model outputs a predicted response for $c_{t+1}$ without further parameter updates. The corresponding true label is $r_{t+1}$.

### 2.2 A Generic Latent Variable Model for ICL

To interpret ICL, Zhang et al. (2025) define a function $f$ that maps a query, a latent variable, and noise to a response:

$$r_i = f(c_i, h_i, \zeta_i), \tag{1}$$

where $h_i$ a latent variable connecting $c_i$ and $r_i$, and $\zeta_i$ are independently and identically distributed random noise for each $i \in [T]$. They also introduce a hidden concept $Z$, from the space $\mathcal{Z}$. The hidden concept represents a shared property across the prompt examples. Based on this concept, the evolution of $h_i$ follows the stochastic process:

$$P(h_i \mid c_i, \{c_s, r_s, h_s\}_{s<i}) = H_{Z^\star}(\{h_s\}_{s<i}, \epsilon_i), \quad (2)$$

for some function $H_{Z^\star}$ and exogenous noises $\{\epsilon_i\}_{i \in [T]}$. This indicates that the distribution of $h_i$ depends on both the hidden concept $Z^\star$ and the prior observations, while it is independent of $c_i$.

The model in (1) assumes that the hidden concept $Z^\star$ implicitly determines the transition of the conditional distribution $P(r_i = \cdot \mid c_i)$ by affecting the evolution of the latent variables $\{h_i\}_{i \in [T]}$, without making any assumptions on the distribution of $c_i$.

*Remark* 2.1. The model is general and encompasses previous models, such as the hidden Markov chain (Xie et al., 2022) and the causal graph (Wang et al., 2023b).

Then Zhang et al. (2025) formulate ICL as follows.

**Lemma 2.1** (Proposition 4.1 of (Zhang et al., 2025)). *Under the model in* (1), *it holds*

$$P(r_{i+1} = \cdot \mid P_i) = \int_{\mathcal{Z}} \mathrm{d}Z \cdot P(r_{i+1} = \cdot \mid P_i, Z) \cdot P(Z \mid \mathcal{D}_i).$$

*Proof.* Please see Appendix C.1 for a detailed proof. $\square$

## 2.3 Memory Reshaping in Modern Hopfield Model

In this part, we introduce modern Hopfield models and review their equivalence to transformer attention mechanisms. We then highlight how memory reshaping in modern Hopfield models corresponds to training transformer attentions.

**Modern Hopfield Model.** Let $x \in \mathbb{R}^d$ represent the query pattern, and $\Xi := [\xi_1, \cdots, \xi_M] \in \mathbb{R}^{d \times M}$ denote the $M$ stored memory patterns. The objective of modern Hopfield model is to encode memory patterns $\Xi$ into its energy landscape and retrieve a memory pattern $\xi_\mu$ based on a given query $x$. Following Ramsauer et al. (2020), the energy function for modern Hopfield model is

$$E(x) = -\mathrm{lse}(\beta, \Xi^\top x) + \frac{1}{2} x^\top x, \quad (3)$$

where $\beta$ is a scaling factor, and $\mathrm{lse}(\beta, \Xi^\top x)$ is the log-sum-exp function defined by

$$\mathrm{lse}(\beta, \Xi^\top x) := \frac{1}{\beta} \log \left( \sum_{i=1}^{M} \exp\left(\beta \xi_i^\top x\right) \right).$$

With the energy function (3), the retrieval dynamics is

$$\mathcal{T}(x_t) := \Xi \,\mathrm{Softmax}\left(\beta \Xi^\top x_t\right) = x_{t+1}. \quad (4)$$

Recent studies establish a formal equivalence between the transformer's attention mechanism and the update dynamics of a modern Hopfield network (Hu et al., 2024b; Wu et al., 2024b; Ramsauer et al., 2020).

*Remark* 2.2. This guarantees that the memories $\{\xi_\mu\}_{\mu \in [M]}$ coincide with the stationary points of $E(x)$, because (4) arises from the stationary condition of the Convex–Concave Procedure (CCCP) (Yuille & Rangarajan, 2001).

**Memory Reshaping.** Memory reshaping in modern Hopfield models refers to modifying the Hopfield energy landscape by relocating its local minima, which encode stored memories. A simple approach to achieve this is by applying a linear transformation to the memory patterns, as demonstrated by Wu et al. (2024a, Eqn. (2.1)). They show that a more uniform distribution of memory representations (i.e., local minima) reduces the number of metastable states in the energy function, thereby enhancing memory capacity and reducing memory confusion. In our work, we analyze an exactly solvable one-layer attention model, and interpret ICL as memory reshaping in the modern Hopfield model from a conditional memory set (in-context examples).

## 2.4 Related Work: In-Context Learning

Several works aim to enhance understanding of ICL from both experimental and theoretical perspectives.

**Empirical Analyses.** Empirically, Zhang et al. (2024) and Min et al. (2022) investigate how covariance shifts affect ICL performance. Yoo et al. (2022) demonstrate that the correctness of input-label mappings is critical for downstream ICL success. Further, Liu et al. (2022) and Rubin et al. (2022) find that increased similarity between prompt examples and test instances enhances ICL performance.

**Theoretical Perspectives.** Theoretically, several studies explain ICL in attention-based models from the gradient descent viewpoint (Von Oswald et al., 2023; Mahankali et al., 2023; Akyürek et al., 2022). In addition, Li et al. (2023) frame ICL as an algorithm learning problem and provide generalization bounds, while Bai et al. (2023) show that ICL performs an implicit algorithm selection based on input sequences. A different line of work adopts a Bayesian framework: Wies et al. (2024), Wang et al. (2023a), Jiang (2023), and Xie et al. (2022) all use Bayesian models to formalize ICL. Notably, Zhang et al. (2025) propose a Bayesian model averaging theory to interpret ICL. We incorporate our proposed in-context memory reshaping phenomenon into their theory. As a result, we provide an exactly solvable example for interpreting ICL with one-layer attention models and complement their results. Further extending the Bayesian

perspective, Han et al. (2024) propose an encoding-decoding mechanism for ICL.

A parallel line of work interprets transformers and ICL through the lens of associative memory. Jiang et al. (2024b) examine context hijacking and show that retrieval of facts in large language models is fragile via an associative memory model. Building on this, Wang & Sato (2024) analyze the interaction between global and in-context knowledge to illustrate how large language models leverage global knowledge in ICL tasks. Bietti et al. (2023) also study ICL from an associative memory perspective. They focus on two-layer transformers, and find that single-layer transformers cannot perform ICL, while two-layer models develop induction heads and succeed. However, their analysis is based on bigram data and does not address the generalization error reduction of ICL. Finally, Cabannes et al. (2024) characterize transformer mechanisms as associative memories and derive scaling laws for large language models. They emphasize scaling behavior, while our work concentrates on the mechanism of ICL itself.

## 3 Main Theory

In this section, we provide an exactly solvable example for interpreting in-context learning with one-layer attention models. First, we derive a memory reshaping formulation of the one-layer attention model under ICL (Section 3.1). We then integrate the proposed in-context memory reshaping into the existing Bayesian model averaging theory of ICL proposed by Zhang et al. (2025), thereby complementing their results and highlighting the advantages of our perspective (Section 3.2). Finally, we characterize how in-context examples "guide" predictions in the Gaussian linear regression and recover the known $\epsilon$-stability generalization bound of the ICL for the one-layer attention model (Section 3.3).

### 3.1 Memory Reshaping for In-Context Learning

In this part, we consider each in-context example as a query to the Hopfield model. Then we show that the sequence of examples induces an effective reshaping of the energy landscape. Specifically, we analyze a one-layer attention model, and illustrate the in-context memory reshaping process by comparing the attention mechanism with and without in-context prompt examples. As a result, we derive an explicit formulation of memory reshaping as a linear transformation of both key and value matrices determined by the prompt.

**With Prompt $P_t = (C_t, R_t, c_{t+1})$.** We consider the case when we have prompt examples for ICL. To align with practice, we map query $c_i \in \mathbb{R}^{d_c}$ to key $k_i \in \mathbb{R}^{d_k}$ by a pre-trained linear layer $k^\star$. Specifically, $k^\star$ is the linear transformation by the pre-trained $W_K^\star$ matrix in the attention head. Then, for any given $C_t$, we have $K_t = [k_1, \cdots, k_t] \in \mathbb{R}^{d_k \times t}$. Similarly, we map response $r_i \in \mathbb{R}^{d_r}$ to $v_i \in \mathbb{R}^{d_v}$

by another pre-trained linear layer $v^\star$ with $v^\star$ being a linear transformation by the pre-trained $W_V^\star$ matrix. Then, for any $R_t$, we have $V_t = [v_1, \cdots, v_t] \in \mathbb{R}^{d_v \times t}$. To calculate the prediction $v_{t+1}$ for the *test query* $c_{t+1}$, we follow (Zhang et al., 2025) and use

$$v_{t+1} = V_t \operatorname{Softmax}\left(K_t^\top q_{t+1}\right), \text{ where } q_{t+1} = k^\star\left(c_{t+1}\right). \tag{5}$$

*Remark* 3.1. Notice that (5) does not represent the actual attention in the LLM. Instead, it serves as a *proxy* attention mechanism for ICL, based on the Bayesian model averaging formulation in Zhang et al. (2025). It provides analytical feasibility for characterizing ICL. By proxy, we refer to the fact that it is an effective attention mechanism derived by framing the ICL as a conditional mean estimation problem. The resulting estimator approximates (5) under a large $t$ limit (Zhang et al., 2025).

**With Prompt $P_0 = (c_{t+1})$.** To align with the expression of the attention mechanism in (5) and to characterize the effects of ICL, we need an equivalent formulation of (5) for scenarios where no in-context prompt examples are included in the input, i.e., $P_0 = (c_{t+1})$. We achieve this by two steps:

First, we define (5) with only test query $c_{t+1}$ as

$$v_{t+1} := W_V^\star c_{t+1}. \tag{6}$$

Second, we prepend $P_0$ with $t$ ghost examples (denoted by $\{\widetilde{c}_i, \widetilde{r}_i\}_{i \in [t]}$) such that "(5) with the ghost prompt $\widetilde{P}_t := (\{\widetilde{c}_i, \widetilde{r}_i\}_{i \in [t]}, c_{t+1})$" has the same output as "(5) with only test query $c_{t+1}$", i.e., (6). We construct such ghost examples $\{\widetilde{c}_i, \widetilde{r}_i\}_{i \in [t]}$ by formulating an optimization problem with parameters $\{\widetilde{c}_i, \widetilde{r}_i\}_{i \in [t]}$. The objective of this optimization problem is to measure the discrepancy between outputs generated with $P_0$ and $\widetilde{P}_t$. Refer to experiments in Section 4.2. The solution to this problem gives the $t$ ghost examples. An example of this type of problem is prompt tuning, notable for its universality (Wang et al., 2024).

*Remark* 3.2 (Ghost Example). We use ghost examples to substitute the real examples in ICL, allowing us to compare scenarios with and without in-context prompt examples while maintaining the same input structure. This setup highlights how the attention mechanism facilitates ICL.

Third, we denote the generated examples as

$$\widetilde{C}_t = [\widetilde{c}_1, \cdots, \widetilde{c}_t], \ \widetilde{R}_t = [\widetilde{r}_1, \cdots, \widetilde{r}_t],$$

where $\widetilde{c}_i \in \mathbb{R}^{d_c}$, $\widetilde{r}_i \in \mathbb{R}^{d_r}$. Using $\widetilde{k}_i = k^\star(\widetilde{c}_i)$, $\widetilde{v}_i = v^\star(\widetilde{r}_i)$, we have

$$\widetilde{K}_t = \left[\widetilde{k}_1, \cdots, \widetilde{k}_t\right], \ \widetilde{V}_t = [\widetilde{v}_1, \cdots, \widetilde{v}_t].$$

To calculate $\widetilde{v}_{t+1}$ for query $c_{t+1}$, we use

$$\widetilde{v}_{t+1} = \widetilde{V}_t \operatorname{Softmax}\left(\widetilde{K}_t^\top q_{t+1}\right). \tag{7}$$

***In-Context* Memory Reshaping.** Comparing (5) and (7), we interpret the two cases as forward passes of an equivalent attention head with reshaped pre-trained attention weight matrices. Under this perspective, in-context learning corresponds to applying linear transformations to $\widetilde{K}_t$, $\widetilde{V}_t$ using matrices $W_1 \in \mathbb{R}^{d_v \times d_v}$, $W_2 \in \mathbb{R}^{d_k \times d_k}$ respectively, i.e.,

$$K_t = W_1 \widetilde{K}_t, \ V_t = W_2 \widetilde{V}_t. \tag{8}$$

$\widetilde{K}_t$ and $\widetilde{V}_t$ are usually rectangular singular matrixs (Feng et al., 2022). Thus, we use Moore–Penrose pseudoinverse to denote the solution to (8). Specifically, we obtain:

$$W_1 = K_t \widetilde{K}_t^\dagger, \ W_2 = V_t \widetilde{V}_t^\dagger.$$

In this way, the ICL reshapes $(k^\star, v^\star)$ to $(W_1 \circ k^\star, W_2 \circ v^\star)$ through the linear transformations, where $\circ$ denotes the function composition.

*Remark* 3.3. In (8), we assume a linear transformation from $\widetilde{K}_t$ to $K_t$, and from $\widetilde{V}_t$ to $V_t$. However, this approach breaks down when $\widetilde{K}_t$ and $\widetilde{V}_t$ are low-rank, as they lack the capacity to express full-rank transformations. In such cases, using the pseudoinverse only minimizes the approximation error but does not yield an exact reconstruction.

We now incorporate the specific form of the attention mechanism to compute $W_1$ and $W_2$, using the mappings $k^\star(c_i) = W_K^\star c_i$, $v^\star(r_i) = W_V^\star r_i$. Under this formulation, we show that ICL is equivalent to applying the following memory reshaping on the memory set, i.e., the key and value matrices:

$$W_K^\star \to W_K^\star C_t \left(W_K^\star \widetilde{C}_t\right)^\dagger W_K^\star,$$
$$W_V^\star \to W_V^\star R_t \left(W_V^\star \widetilde{R}_t\right)^\dagger W_V^\star. \tag{9}$$

Therefore, the ICL prediction in (5) with $t$ in-context examples becomes:

$$v_{t+1} = \widetilde{W}_V^\star R_t \cdot \text{Softmax}\left(\left(\widetilde{W}_K^\star C_t\right)^\top W_K^\star c_{t+1}\right), \tag{10}$$

where

$$\widetilde{W}_K^\star := W_K^\star C_t (W_K^\star \widetilde{C}_t)^\dagger W_K^\star,$$
$$\widetilde{W}_V^\star := W_V^\star R_t (W_V^\star \widetilde{R}_t)^\dagger W_V^\star.$$

In this way, we provide the first exactly solvable example for interpreting ICL with one-layer attention models, by deriving the in-context memory reshaping formulation based on linear transformations.

## 3.2 Memory Reshaping as a Complement to Bayesian Model Averaging in ICL (Zhang et al., 2025)

In this part, we integrate our proposed in-context memory reshaping phenomenon into the existing Bayesian model

averaging theory of ICL (Zhang et al., 2025), and provide an exactly solvable example to complement the results in (Zhang et al., 2024). We employ the same Gaussian linear model to clearly illustrate how the attention mechanism performs ICL following (Zhang et al., 2025):

$$v_{t+1} = Z^\star \phi\left(k^\star\left(c_{t+1}\right)\right) + \eta_{t+1}, \tag{11}$$

where $\phi : \mathbb{R}^{d_k} \to \mathbb{R}^{d_\phi}$ denotes the feature mapping in reproducing kernel Hilbert space, $Z^\star \in \mathbb{R}^{d_v \times d_\phi}$ denotes the hidden concept, and $\eta_{t+1} \in N(0, \sigma_1^2 I_{d_v})$ with $\sigma_1 > 0$ are independently and identically distributed Gaussian noises. Besides, we assume that $Z^\star$ follows Gaussian distribution $N(0, \lambda^2 I_{d_v \times d_\phi})$. By incorporating the specific attention mapping $k^\star$ and the memory reshaping formulation (9), we obtain the following for the Gaussian linear model:

$$v_{t+1} = Z^\star \phi\left(W_K^\star C_t (W_K^\star \widetilde{C}_t)^\dagger W_K^\star c_{t+1}\right) + \eta_{t+1}. \tag{12}$$

By Lemma 2.1, we have the following result:

**Proposition 3.1** (Bayesian Model Averaging in Gaussian Linear Model, Modified from (Zhang et al., 2024))**.** *By* (12), *Lemma* 2.1 *implies*

$$P(v_{t+1} \mid P_t) = \int_{\mathcal{Z}} dZ \cdot P(v_{t+1} \mid Z, q_{t+1}) P(Z \mid \mathcal{D}_t)$$
$$\propto \exp\left(\frac{-\left\|v_{t+1} - \bar{Z}_t \phi(q_{t+1})\right\|_2^2}{2\Sigma_t}\right), \tag{13}$$

*where*

$$K_t = W_K^\star C_t (W_K^\star \widetilde{C}_t)^\dagger W_K^\star C_t,$$
$$V_t = W_V^\star R_t (W_V^\star \widetilde{R}_t)^\dagger W_V^\star R_t,$$
$$q_{t+1} = W_K^\star C_t (W_K^\star \widetilde{C}_t)^\dagger W_K^\star c_{t+1},$$
$$\bar{Z}_t = V_t (\sigma_1^2 \lambda^{-2} I + \phi(K_t) \phi(K_t)^\top)^{-1} \phi(K_t)^\top,$$
$$\Sigma_{Z,t} = \lambda^{-2} I + \sigma_1^{-2} \phi(K_t) \phi(K_t)^\top,$$
$$\Sigma_t = \sigma_1^2 I + \phi(q_{t+1})^\top \Sigma_{Z,t}^{-1} \phi(q_{t+1}).$$

*Proof.* Please see Appendix C.2 for a detailed proof. □

*Remark* 3.4. In Proposition 3.1, $\bar{Z}_t \phi(q_{t+1})$ measures the similarity between the query and keys. This is similar to the attention mechanism. However, the normalization of similarity is not softmax. This motivates us to define a new structure of attention and explore the relationship between the newly defined attention and the original one.

*Remark* 3.5. $\bar{Z}_t \phi(q_{t+1})$ represents the weighted average of values $\{v_i\}_{i\in[t]}$ in $V_t$. As $t$ increases, $\bar{Z}_t$ converges to a more stable and robust hidden state.

To connect with the actual transformer attention mechanism, we restate the variant of attention (Bayesian model averaging attention) defined in (Zhang et al., 2025).

**Definition 3.1** (Bayesian Model Averaging Attention (Zhang et al., 2025)). $\bar{Z}_t \phi(q_{t+1})$ measures the similarity between the query and the keys, similar to the role of the attention mechanism. We restate the newly defined variant of the attention mechanism in (Zhang et al., 2025) as follows:

$$A(q, K_t, V_t) = V_t \left(\sigma_1^2 \lambda^{-2} I + \mathsf{K}(K_t^\top, K_t^\top)\right)^{-1} \mathsf{K}(K_t^\top, q), \tag{14}$$

where $\mathsf{K}(\cdot, \cdot)$ denotes the kernel function of the reproducing kernel Hilbert space induced by $\phi$.

Next, we restate the results in (Zhang et al., 2025) to show that the attention in (14) is equivalent to softmax attention.

**Assumption 3.1.** Assume key-value pairs $\{k_t, v_t\}_{t=1}^T$ are independently and identically distributed. Further, assume that each vector satisfies $\|k_i\|_{2,2} = \|v_i\|_{2,2} = 1$, and that any query $q \in \mathbb{R}^{d_k}$ also satisfies $\|q\|_{2,2} = 1$.

*Remark* 3.6. Assumption 3.1 further assumes that $\{c_t, r_t\}_{t=1}^T$ are independently and identically distributed. However, this assumption may not hold for the ghost examples due to dependencies introduced by the joint optimization (Section 3.1). To mitigate this issue, we propose introducing a covariance-based regularization term that encourages the covariance matrix of the ghost examples to approximate the identity matrix.

**Lemma 3.1** (Bayesian Model Averaging Attention Approximates Softmax Attention, Proposition 4.3 of (Zhang et al., 2025) ). *For any absolute constant $C > 0$, we have:*

$$\lim_{t \to \infty} A(q, K_t, V_t) = C \cdot \lim_{t \to \infty} V_t \operatorname{Softmax}(K_t^\top q).$$

*Proof.* Please see (Zhang et al., 2025, Appendix E.4) for a detailed proof. □

*Remark* 3.7. That is, when the number of prompt examples goes to infinity, the attention in Definition 3.1 is equal to Softmax-attention up to a constant.

*Remark* 3.8. We use memory reshaping to demonstrate how prompt examples influence model performance, thereby complementing the results in (Zhang et al., 2025). We interpret this effect as a linear transformation applied to the pre-trained attention weights.

### 3.3 Memory Reshaping for Characterizing Gaussian Linear Model and ICL Generalization Bound

In this part, we use our proposed in-context memory reshaping phenomenon to characterize how in-context examples shape predictions in the Gaussian linear regression and recover the known $\epsilon$-stability generalization bound of the ICL for the one-layer attention model.

**Bayes-Optimal Prediction of Gaussian Linear Regression.** For the Gaussian linear model in (11), we characterize the bayes-optimal prediction using Proposition 3.1:

$$\bar{Z}_t = V_t(\sigma_1^2 \lambda^{-2} I + \phi(K_t)\phi(K_t)^\top)^{-1}\phi(K_t)^\top,$$

where

$$K_t = W_K^\star C_t (W_K^\star \widetilde{C}_t)^\dagger W_K^\star C_t,$$
$$V_t = W_V^\star R_t (W_V^\star \widetilde{R}_t)^\dagger W_V^\star R_t.$$

This formulation shows how prompts steer predictions toward the Bayes-optimal solution by dynamically re-weighting stored memories. This serves as an explicit characterization of how prompts "guide" predictions.

**In-Context Learning Generalization Bound.** Let $r_{t+1}$ be the true response corresponding to the query $c_{t+1}$. Then the true target for prediction in (10) is $\widetilde{W}_V^\star r_{t+1}$. We define the loss function for predicting $v_{t+1}$ as $L(\mathcal{D}_t, c_{t+1}) = \mathcal{L}(v_{t+1}, \widetilde{W}_V^\star r_{t+1})$, where $\mathcal{L}$ is a task-specific loss function, e.g., mean squared error loss. The generalization error of in-context learning with prompt $\mathcal{D}_t$ and test query $c_{t+1}$ is $R(\mathcal{D}_t) = \mathbb{E}_{c_{t+1}}[L(\mathcal{D}_t, c_{t+1})]$. The empirical risk is $R_{\text{emp}}(\mathcal{D}_t) = 1/t \cdot \sum_{i=1}^t L(\mathcal{D}_t, c_i)$, i.e., the average loss when each $c_i$ in the prompt acts as the test query.

We make the following assumptions on the boundedness of the weight matrices and the inputs, as well as the Lipschitz continuity of the loss function $\mathcal{L}$.

**Assumption 3.2** (Norm Bounds). Assume the following boundedness conditions hold:

$$\|(\widetilde{W}_K^\star)^\top W_K^\star\|_{2,2} \le \Gamma, \ \|\widetilde{W}_V^\star\|_{2,2} \le 1.$$

Further assume that $c_i$ and $r_i$ lie in the unit ball, i.e.,

$$\|C_t\|_{2,\infty}, \ \|R_t\|_{2,\infty}, \ \|c_{t+1}\|_2 \le 1.$$

**Assumption 3.3** ($M$-Lipschitz of $\mathcal{L}$).

$$|\mathcal{L}(v_1, v_{\text{true}}) - \mathcal{L}(v_2, v_{\text{true}})| \le M\|v_1 - v_2\|_2.$$

*Remark* 3.9. The three assumptions are reasonable. Assumption 3.1 and Assumption 3.2 conform to most real-world situations due to the layer normalization in the Transformer architecture and the regularization loss term used in the model training (Li et al., 2023; Xiong et al., 2020; Lewkowycz & Gur-Ari, 2020). These mechanisms help ensure numerical stability and prevent overfitting, making the assumptions practically applicable. In addition, Assumption 3.3 is a common assumption in machine learning theory (Li et al., 2023; Edelman et al., 2022).

**Theorem 3.1** (Generalization Bound of ICL for One-Layer Attention Model). *The generalization bound of ICL for the one-layer attention model is as follows:*

$$\mathbb{E}_{\mathcal{D}_t}[R(\mathcal{D}_t) - R_{emp}(\mathcal{D}_t)] \le \frac{MC_0(4\Gamma + 1)e^{2\Gamma}}{t},$$

*where $C_0 > 0$ is an absolute constant.*

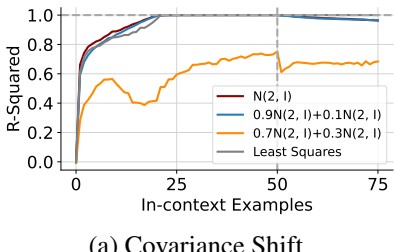

(a) Covariance Shift

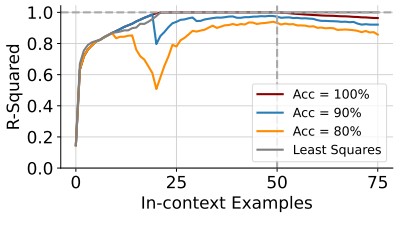

(b) Response's Accuracy

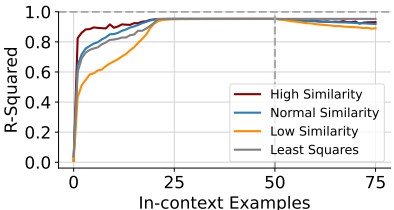

(c) Similarity with *Test Query*

*Figure 1.* **Behaviors of In-Context Learning Measured by R-Squared in Linear Regression Problem.** The true response to the query is generated by a linear model. (a) shows the performance of ICL under different covariance shifts. (b) shows the performance of ICL under different accuracies of responses in prompts. (c) shows the performance of ICL under different similarities between prompts and the *test query*. "Least Squares" denotes the result using the least squares method directly. Unless otherwise specified, the baseline setting uses the query distribution as $N(-2, I)$, the response's accuracy as $100\%$, and the similarity level with the *test query* as normal.

*Proof Sketch.* Our proof utilizes the connection between $\epsilon$-stability andthe generalization bound. Firstly, we view ICL as the algorithm described in Li et al. (2023). Secondly, we calculate the $\epsilon$-stability of the one-layer attention model for ICL (Li et al., 2023). Finally, we derive the generalization bound of the one-layer attention model for ICL, leveraging the connection between $\epsilon$-stability and generalization bound (Bousquet & Elisseeff, 2002). Please see Appendix C.3 for a detailed proof. □

*Remark* 3.10. The generalization bound recovers the known $\epsilon$-stability bound for ICL (Li et al., 2023) within our one-layer attention setting. This confirms that models following our memory retrieval interpretation achieve the same established generalization guarantees for ICL. Furthermore, the generalization bound supports the intuition that ICL performance improves with more reliable in-context examples.

## 4 Experimental Studies

In this section, we use the *in-context memory reshaping* phenomenon to interpret and design experiments to validate three key properties of ICL: the model's sensitivity to (i) covariance shifts, (ii) response accuracy, and (iii) the similarity between prompts and test examples. Furthermore, our experimental results validate that the generalization bound improves with more prompt examples in Theorem 3.1. At last, we demonstrate our ability to construct $t$ ghost examples $\{\widetilde{c}_i, \widetilde{r}_i\}_{i \in [t]}$ with prompt tuning.

Our experiments cover both synthetic and real-world tasks: (i) Following (Garg et al., 2022), we use the GPT-2 model on synthetic problems, including linear regression, decision trees, and 2-layer neural networks; (ii) We use the GPT-J model on a sentiment classification task. Take the linear regression setting with GPT-2 as an example, we define the target function as $f(x) = \beta^T x, \beta \sim \mathbb{R}^d$, where $d = 20$. The distribution of $x \in \mathbb{R}^d$ is from a Gaussian Mixture model $\omega_1 N(-2, I) + \omega_2 N(2, I)$, where $\omega_1 = 1, \omega_2 = 0$ in the pre-training. The pre-training process refers to the method in (Garg et al., 2022), and please see Appendix D.1

for the details. For other tasks, we leave more details in Appendices D.5 and D.6.

### 4.1 Memory Reshaping for Three Behaviors of ICL

Here we consider three behaviors of in-context learning: sensitivity to (i) covariance shifts, (ii) response accuracy, and (iii) the similarity between prompts and *test query*.

*Table 1.* **Sensitivity to Covariance Shifts: R-Squared under Different Test Data Distributions.** 15, 30, 45, 60, 75 denote the in-context example size. The training data distribution is $N(-2, I)$, and in-context sample size is 50. "Least Squares" denotes the baseline by least squares regression.

| Test Distribution | 15 | 30 | 45 | 60 | 75 |
|---|---|---|---|---|---|
| Least Squares | 0.8811 | 1.0 | 1.0 | 1.0 | 1.0 |
| $N(-2, I)$ | 0.9366 | 0.9998 | **0.9999** | 0.9838 | 0.9641 |
| $0.9N(-2, I) + 0.1N(2, I)$ | 0.9202 | 0.9997 | **0.9998** | 0.9797 | 0.9677 |
| $0.7N(-2, I) + 0.3N(2, I)$ | 0.4043 | 0.6613 | **0.7337** | 0.6678 | 0.6792 |

*Table 2.* **Sensitivity to Response's Accuracy: R-Squared under Different Response Accuracies of In-Context Examples.** 15, 30, 45, 60, 75 denote the in-context example size. The in-context sample size in the training data is 50. "Least Squares" denotes the baseline by least squares regression.

| Response Accuracy | 15 | 30 | 45 | 60 | 75 |
|---|---|---|---|---|---|
| Least Squares | 0.8811 | 1.0 | 1.0 | 1.0 | 1.0 |
| 100% | 0.9144 | 0.9998 | **0.9999** | 0.9800 | 0.9578 |
| 90% | 0.9144 | 0.9614 | **0.9677** | 0.9507 | 0.9075 |
| 80% | 0.8266 | 0.8607 | **0.9230** | 0.8955 | 0.8512 |

**Sensitivity to Covariance Shifts.** An important factor affecting the effectiveness of in-context learning is the difference between the query distribution in the pre-training dataset and the testing dataset, i.e., covariance shifts (Zhang et al., 2024; Min et al., 2022). In-context learning performs worse when the difference is larger.

We explain covariance shifts through memory reshaping as follows: When there are no covariance shifts, the reshaped

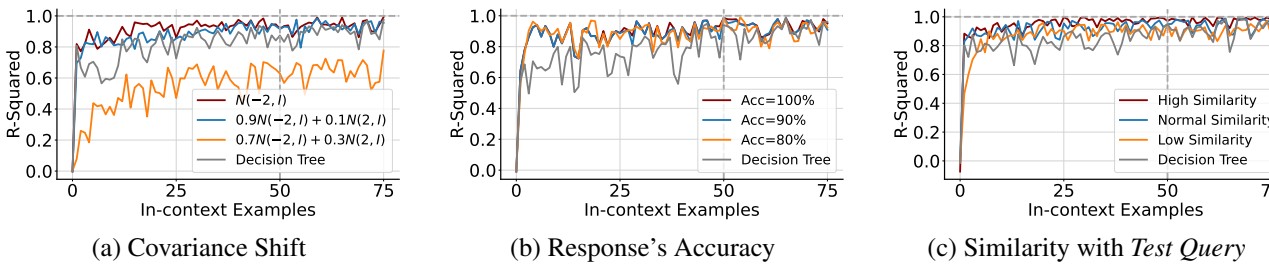

(a) Covariance Shift     (b) Response's Accuracy     (c) Similarity with *Test Query*

*Figure 2.* **Behaviors of In-Context Learning Measured by R-Squared in Decision Tree Problem.** The true response to the query is generated by a decision tree. (a) shows the performance of ICL under different covariance shifts. (b) shows the performance of ICL under different accuracies of responses in prompts. (c) shows the performance of ICL under different similarities between prompts and the *test query*. "Decision Tree" denotes the baseline, i.e., using the decision tree method directly. Unless otherwise specified, the baseline setting uses the query distribution as $N(-2, I)$, the response's accuracy as $100\%$, and the similarity level with the *test query* as normal.

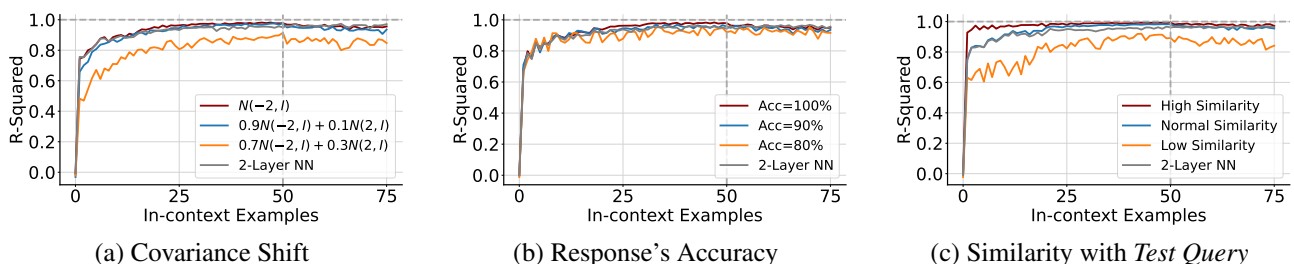

(a) Covariance Shift     (b) Response's Accuracy     (c) Similarity with *Test Query*

*Figure 3.* **Behaviors of In-Context Learning Measured by R-Squared in 2-Layer Neural Network Problem.** The true response to the query is generated by a 2-layer feed-forward neural network. (a) shows the performance of ICL under different covariance shifts. (b) shows the performance of ICL under different accuracies of responses in prompts. (c) shows the performance of ICL under different similarities between prompts and the *test query*. "2-Layer NN" denotes the baseline, i.e., using the 2-layer neural network trained with prompt examples for 100 gradient descent steps. Unless otherwise specified, the baseline setting uses the query distribution as $N(-2, I)$, the response's accuracy as $100\%$, and the similarity level with the *test query* as normal.

memory's distribution matches the initial memory distribution after pre-training. With distribution shifts, the distribution of reshaped memory differs from the initial memory's distribution. A larger shift in covariance means a greater difference between the distribution before and after reshaping. The model struggles to reshape the memory perfectly with a different memory distribution. The difficulty of reshaping increases as the distributions become more different.

We conduct the experiments to test the model's performance with different distributions of queries in the testing dataset. We use three different Gaussian mixture distribution $\omega_1 N(-2, I) + \omega_2 N(2, I)$ with different $\omega_1, \omega_2$: (i). $\omega_1 = 1, \omega_2 = 0$, (ii). $\omega_1 = 0.9, \omega_2 = 0.1$, (iii). $\omega_1 = 0.7, \omega_2 = 0.3$. Here the distribution in the first setting matches the distribution in pre-training. During testing, we generate samples similar to those used in pre-training. We extend the prompt length from 50 to 75 to evaluate the performance of ICL with longer prompts than those used in pre-training. For each in-context length $j \in [75]$, we measure performance using the R-squared between the estimation and the true value. See Appendix D.2 for the details.

We show the results in Figure 1 (a) and Table 1. The results verify that in-context learning performs worse with larger covariance shifts in the testing dataset.

*Remark* 4.1. We find that performance decreases when the prompt length exceeds the pre-training length (i.e., 50), a well-known issue. We believe this is due to the absolute positional encodings, as noted in Zhang et al. (2024).

*Table 3.* **Sensitivity to Similarity between Prompts and *Test Query*: R-Squared Value under Different Similarity between Prompts and *Test Query*.** $15, 30, 45, 60, 75$ denotes the in-context example size. The in-context sample size in the training data is 50. "Least Squares" denotes the baseline by least squares regression.

| Similarity Degree | 15 | 30 | 45 | 60 | 75 |
|---|---|---|---|---|---|
| Least Squares | 0.8811 | 1.0 | 1.0 | 1.0 | 1.0 |
| *Best* | 0.9563 | 0.9997 | **0.9998** | 0.9899 | 0.9756 |
| *Normal* | 0.9366 | 0.9998 | **0.9999** | 0.9838 | 0.9641 |
| *Worst* | 0.7704 | 0.9985 | **0.9993** | 0.9673 | 0.9290 |

**Sensitivity to Response's Accuracy.** Another factor influencing the performance of ICL is the response format (Yoo et al., 2022). Replacing the response set with a random set reduces the performance of ordinary auto-regressive LLMs. We explain this using memory reshaping as follows: The prompt response contributes to the equivalent memory reshaping process. The wrong response set misleads the re-

shaping process. Therefore, accurate responses in prompts are crucial for effective ICL.

We conduct the following experiments to test the model performance with varying response accuracies. We use the same Gaussian mixture distribution as in pre-training, i.e., $\omega_1 N(-2, I) + \omega_2 N(2, I)$, where $\omega_1 = 1, \omega_2 = 0$. We generate samples with different accuracy of response, see Appendix D.3 for the details. We use the same method to calculate the R-squared as Section 4.1. Here we visualize the performance with accuracies of $100\%$ and $80\%$ in Figure 1 (b) and show partial results in Table 2. The results confirm that ICL performs worse when the accuracy of the response in prompts is lower.

**Sensitivity to Similarity between Prompts and *Test Query*.** The similarity between prompts and the *test query* is also one important factor to influence the performance of in-context learning (Liu et al., 2022; Rubin et al., 2022). The higher similarity generates better in-context learning performance. We give an explanation about this by memory reshaping as follows: Similar queries have similar responses. For example, in a linear regression problem, the difference between the responses for similar queries is small. in-context learning reshapes $\widetilde{V}_t$ to $W_2\widetilde{V}_t$ as in (8). Compared with $\widetilde{V}_t$, the values in $W_2\widetilde{V}_t$ are much more approximate to the ideal value of *test query*. Therefore, the higher similarity between prompts and *test query* benefits the in-context learning.

We conduct the following experiments to test the model performance with different similarities. We use the same Gaussian mixture distribution as the pre-training, i.e., $N(-2, I)$. We use the cosine similarity between different queries as the similarity measure, and generate prompts with different similarity as the *test query*, see Appendix D.4 for the details. We calculate the R-squared as Section 4.1. Here we visualize the performance in three settings in Figure 1 (c): high, normal, and low similarity, and show partial results in Table 3. The results verify that the in-context learning performs better when the similarity is higher.

**More Complex Datasets and Models.** In this part, we extend the experiment from linear regression to a decision tree and 2-layer neural network. We use the decision tree and 2-layer neural network to act as the function $f$. We leave the details about the experiment setting in Appendix D.5. We show the results of the decision tree in Figure 2 and 2-layer neural network in Figure 3.

Furthermore, we extend the analysis from GPT-2 to the more capable GPT-J model and evaluate it on the real-world "TweetEval: Hate Speech Detection" dataset (Nagel, 2016) for a sentiment classification task. We show the detailed experimental settings and results in Appendix D.6. The results follow a similar pattern to the linear regression setting.

**Validation of Generalization Bound.** We also validate the generalization bound in Theorem 3.1. The results in Figures 1 to 3 show that the R-squared value increases when the number of in-context examples is below 50—the same number used during pre-training. This supports the our observation that the generalization bound improves with more prompt examples.

### 4.2 Ghost Example Construction

We conduct experiments to demonstrate our ability to construct $t$ ghost examples $\{\widetilde{c}_i, \widetilde{r}_i\}_{i\in[t]}$ with prompt tuning. We calculate the squared error between the outputs with $c_{t+1}$ or $\widetilde{P}_t := (\{\widetilde{c}_i, \widetilde{r}_i\}_{i\in[t]}, c_{t+1})$ as the model inputs, where $t \in [75]$. To generate $c_{t+1}$, we use two distributions $\omega_1 N(-2, I) + \omega_2 N(2, I)$, one is the same as the pre-training data distribution, and another is different from it, i.e., $\omega_1 = 1, \omega_2 = 0$ or $\omega_1 = 0.5, \omega_2 = 0.5$. The average mean squared error for all $t \in [75]$ is 0.0003 and 0.001 respectively. See Appendix D.7 for the details.

## 5 Discussion and Conclusion

In this work, we provide an exactly solvable example for interpreting In-Context Learning (ICL) with one-layer attention models as conditional retrieval of dense associative memory models. We interpret ICL as memory reshaping in the modern Hopfield model from a conditional memory set (in-context examples), and propose the *in-context memory reshaping* phenomenon (Section 3.1). We then integrate this phenomenon into the existing Bayesian model averaging theory of ICL (Zhang et al., 2025) to complement their results (Section 3.2). Moreover, under this unique perspective, we not only characterize how in-context examples shape predictions in the Gaussian linear regression case, but also recover the known $\epsilon$-stability generalization bound of the ICL for the one-layer attention model (Section 3.3). We also give explanations for three key behaviors of ICL and validate them through experiments on both synthetic and real datasets (Section 4). These behaviors include sensitivity to covariance shifts, sensitivity to response's accuracy, and sensitivity to the similarity between prompts and *test query*.

Our findings enhance the understanding of ICL and its performance dynamics, paving the way for further advancements in LLM capabilities. By interpreting LLM as an energy-based model, we have three practical guidelines for practitioners: (i) As demonstrated in Section 4.1, selecting prompts similar to the test query enhances in-context learning performance. This aligns with the findings of retrieval-augmented generation (Lewis et al., 2020). (ii) By Theorem 3.1, increasing the number of relevant prompt examples reduces the ICL generalization error. (iii) As stated in Theorem 3.1, a smaller norm bound $\Gamma$ (Assumption 3.2) improves ICL response quality. Furthermore, we discuss the limitations of our work in Appendix A.

## Impact Statement

This research is theoretical and is not expected to have negative social impacts. As outlined in the introduction and related works, the primary goal of this study is to enhance our understanding of the underlying principles of large foundation models from an associative memory perspective.

## Acknowledgments

The authors thank Zhang et al. (2025) for clarifications and helpful feedback; Jiachen Zhao, Mimi Gallagher, Sara Sanchez, Dino Feng and Andrew Chen for enlightening discussions; Zhao Song, Dennis Wu, Jennifer Zhang, Maojiang Su, Hude Liu and Hong-Yu Chen for collaborations on related topics; and Jiayi Wang for facilitating experimental deployments. JH also thanks the Red Maple Family for support. The authors would like to thank the anonymous reviewers and program chairs for constructive comments.

JH is partially supported by the Walter P. Murphy Fellowship. Han Liu is partially supported by NIH R01LM1372201, NSF AST-2421845, Simons Foundation MPS-AI-00010513, AbbVie and Dolby. This research was supported in part through the computational resources and staff contributions provided for the Quest high performance computing facility at Northwestern University which is jointly supported by the Office of the Provost, the Office for Research, and Northwestern University Information Technology. The content is solely the responsibility of the authors and does not necessarily represent the official views of the funding agencies.

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

# Supplementary Material

# A  Limitations

We also have the following four limitations in our work:

- One notable limitation is that the $\mathrm{Softmax}$-attention in (5) does not represent the actual attention in the LLM. Although this effective attention mechanism helps us better understand ICL in the Bayesian model averaging view, to ensure our theory is more aligned with reality, we should consider the actual attention in the LLM.

- In our theory, we only consider the one-layer attention mechanism. We plan to study how a multi-layer attention mechanism performs ICL in future work. We regard the multi-layer transformer as iteratively retrieving memory patterns in the modern Hopfield model through the following four steps: (i) Design an energy function for multi-layer modern Hopfield model: We design an energy function matching the multi-layer transformer (Hoover et al., 2023). The memory pattern needs retrieval after reshaping and iteration. This step is challenging. (ii) Give a clear expression of memory reshaping on model weights: This step is feasible following our framework in Section 3.1. (iii) Extend the Bayesian method to a multi-layer transformer: The Bayesian method in (Zhang et al., 2025) and our work assumes perfect pre-training. We must study perfect pre-training to match real multi-layer transformers. This step is challenging. (iv) Obtain a generalization bound of ICL: This step is feasible following our framework in Appendix C.3. We need to derive the $\epsilon$-stability of the multi-layer transformer.

- Our analysis focuses on predicting a single next token. Traditional autoregressive large language models often use in-context learning to predict multiple tokens. We now describe how to extend our method to this setting. This extension involves iteratively applying the same single-step retrieval mechanism: (i) After generating the first token, append it to the context to form a new query. (ii) Each updated query triggers the memory reshaping of ICL in large language models to predict the next token. (iii) Repeating this process yields the full output sequence. For example, to predict two tokens, the prompt queries and responses are $C_t = [c_1, r_{1,1}, \ldots, c_t, r_{t,1}] \in \mathbb{R}^{d_c \times 2t}$ and $R_t = [r_{1,1}, r_{1,2}, \ldots, r_{t,1}, r_{t,2}] \in \mathbb{R}^{d_r \times 2t}$, with $\mathcal{D}_t := \{c_i, r_{i,1}, r_{i,2}\}_{i=1}^t$. The LLM takes prompt $P_t = (\mathcal{D}_t, c_{t+1})$ and predicts $r_{t+1,1}$ and $r_{t+1,2}$. The joint probability follows $P(r_{i+1,1} = \cdot, r_{i+1,2} = \cdot \mid P_i) = P(r_{i+1,2} = \cdot \mid r_{i+1,1} = \cdot, P_i) \cdot P(r_{i+1,1} = \cdot \mid P_i)$. Our framework in Section 3.1 and Section 3.2 applies to $r_{i+1,1}$. Once we predict $r_{i+1,1}$, we apply the same method to obtain $P(r_{i+1,2} = \cdot \mid r_{i+1,1} = \cdot, P_i)$. For generalization in Section 3.3, the bound for $r_{i+1,1}$ uses the existing framework. For $r_{i+1,2}$, we treat prediction error in $r_{i+1,1}$ as noise to the input. This matches the setting in our $\epsilon$-stability analysis (Lemma C.2). We only adjust a few hyperparameters to extend the bound to $r_{i+1,2}$.

- We do not verify the exact memory reshaping setting in our current experiments. This is challenging. However, we provide a potential solution under an ideal setting, and level it for future work. Following Appendix D, for each $t \in [75]$, we generate 6400 samples with the distribution $N(-2, I)$. Each sample follows the pre-training setup with batch size 1. For the $i$-th sample, we define prompt $P_{i,t} = (C_{i,t}, R_{i,t}, c_{i,t+1})$. We use $c_{i,t+1}$ to construct $t$ ghost examples via prompt tuning (see Section 3.1). We denote the ghost query and response as $\{\widetilde{c}_{i,j}, \widetilde{r}_{i,j}\}_{j \in [t]}$. The model outputs $\widehat{r}_{i,t+1}$ when inputs are $(C_{i,t}, R_{i,t}, c_{i,t+1})$. To simulate memory reshape, we input $(\widetilde{C}_{i,t}, \widetilde{R}_{i,t}, c_{i,t+1})$ and modify attention layers as follows: (i) In each attention layer, we compute $\widetilde{K}_{i,t}$ and $\widetilde{V}_{i,t}$. (ii) We compare with the original $K_{i,t}$, $V_{i,t}$, and solve (8) to get reshape matrices $W_{i,1}$ and $W_{i,2}$. (iii) We apply them as $W_{i,1}\widetilde{K}_{i,t}$ and $W_{i,2}\widetilde{V}_{i,t}$ in the forward pass. This method simulates the effect of memory reshaping. However, implementing and validating this approach is difficult. It requires fine-grained control over internal activations at each layer. It also involves layer-wise matrix inversion and alignment for every sample and timestep, which can introduce instability and noise. Due to these challenges, we leave full implementation and validation to future work. Nonetheless, the setup outlines a feasible path and offers useful guidance for follow-up research.

# B   Notation Table

We summarize our notations in the following table for easy reference.

*Table 4.* Mathematical Notations and Symbols

| Symbol | Description |
|---|---|
| $d_k$ | Dimension of key vectors and query vectors |
| $d_v$ | Dimension of value vectors |
| $d_\phi$ | Dimension of the output of feature mapping $\phi$ |
| $C_t$ | Input query, defined as $C_t = [c_1, \ldots, c_t] \in \mathbb{R}^{d_c \times t}$ |
| $R_t$ | Input response, defined as $R_t = [r_1, \ldots, r_t] \in \mathbb{R}^{d_r \times t}$ |
| $\mathcal{D}_t$ | Input prompt examples pair, defined as $\mathcal{D}_t \coloneqq \{c_i, r_i\}_{i=1}^t$ |
| $P_t$ | Input prompt, defined as $P_t = (\mathcal{D}_t, c_{t+1})$ |
| $k$ | Key vector in $\mathbb{R}^{d_k}$ |
| $K_t$ | Key matrix, defined as $K_t = [k_1, \cdots, k_t] \in \mathbb{R}^{d_k \times t}$ |
| $\phi(\cdot)$ | Kernelized feature mapping: $\mathbb{R}^{d_k} \to \mathbb{R}^{d_\phi}$ |
| $v$ | Value vectors in $\mathbb{R}^{d_v}$ |
| $V_t$ | Value matrix, defined as $V_t = [v_1, \cdots, v_t] \in \mathbb{R}^{d_v \times t}$ |
| $q$ | Query vector in $\mathbb{R}^{d_k}$ |
| $A(q, K_t, V_t)$ | Bayesian model averaging attention defined by Definition 3.1 |
| $Z^\star$ | Hidden concept representing a property shared across all prompt examples. |
| $\bar{Z}_t$ | Posterior Mean of $Z^\star$ given prompt $\mathcal{D}_t$ |
| $h_t$ | Latent variable linking query $c_t$ to response $r_t$ |
| $\mathcal{H}$ | Space of latent variable $h_t$ |
| $\mathcal{Z}$ | Space of hidden concept $Z$ |
| $\Sigma_{Z,t}$ | Posterior covariance of $Z^\star$ given prompt $\mathcal{D}_t$ |
| $\Sigma_t$ | Predicted covariance of $v_{t+1}$ conditional on $P_t$ |
| $\sigma_1$ | Standard deviation of noise in Gaussian linear model |
| $\lambda$ | Prior standard deviation of elements in latent concept $Z^\star$ |
| $W_K^\star$ | Pre-trained key-projection matrix |
| $W_V^\star$ | Pre-trained value-projection matrix |
| $\widetilde{W}_K^\star$ | Reshaped key-projection matrix after memory reshaping |
| $\widetilde{W}_V^\star$ | Reshaped value-projection matrix after memory reshaping |
| $L(\cdot, \cdot)$ | Loss function with respect to $(\mathcal{D}_t, c_{t+1})$ |
| $\mathcal{L}(\cdot, \cdot)$ | Task-specific loss function, e.g., mean squared error |

# C   Proofs of Main Text

In this section, we provide the detailed proof of Lemma 2.1 and Proposition 3.1.

## C.1   Proof of Lemma 2.1

*Proof Sketch.* This proof follows Zhang et al. (2025, Proposition 4.1). It is based on a general latent variable model and uses the law of total probability to derive the desired result. $\square$

*Proof.* Recall that $P_t = (\mathcal{D}_t, c_{t+1})$, where $\mathcal{D}_t$ is the set of prompt examples and $c_{t+1}$ is the test query. $h_t \in \mathcal{H}$ denotes the hidden variable, and $Z \in \mathcal{Z}$ is the hidden concept. Then, we have

$$
\begin{aligned}
P(r_{t+1} \mid P_t) &= \int_{\mathcal{H}} \mathrm{d}h_{t+1} \cdot P(r_{t+1}, h_{t+1} \mid P_t) \\
&= \int_{\mathcal{H}} \mathrm{d}h_{t+1} \cdot P(r_{t+1} \mid P_t, h_{t+1}) P(h_{t+1} \mid P_t) \\
&= \int_{\mathcal{H}} \mathrm{d}h_{t+1} \cdot P(r_{t+1} \mid c_{t+1}, h_{t+1}, \mathcal{D}_t) \cdot P(h_{t+1} \mid c_{t+1}, \mathcal{D}_t) \\
&= \int_{\mathcal{H}} \mathrm{d}h_{t+1} \cdot P(r_{t+1} \mid c_{t+1}, h_{t+1}) \cdot P(h_{t+1} \mid \mathcal{D}_t) \\
&= \int_{\mathcal{H}} \mathrm{d}h_{t+1} \cdot P(r_{t+1} \mid c_{t+1}, h_{t+1}) \cdot \left( \int_{\mathcal{Z}} \mathrm{d}Z \cdot P(h_{t+1} \mid \mathcal{D}_t, Z) \cdot P(Z \mid \mathcal{D}_t) \right) \\
&= \int_{\mathcal{Z}} \mathrm{d}Z \cdot \left( \int_{\mathcal{H}} \mathrm{d}h_{t+1} \cdot P(r_{t+1} \mid c_{t+1}, h_{t+1}) \cdot P(h_{t+1} \mid \mathcal{D}_t, Z) \right) \cdot P(Z \mid \mathcal{D}_t) \\
&= \int_{\mathcal{Z}} \mathrm{d}Z \cdot P(r_{t+1} \mid P_t, Z) \cdot P(Z \mid \mathcal{D}_t),
\end{aligned}
$$

where the first line is by the law of total probability to integrate over $h_{t+1}$, the second line is by factorizing the joint conditional density of $r_{t+1}$ and $h_{t+1}$ given $P_t$, the third line is by substituting $P_t = (\mathcal{D}_t, c_{t+1})$, the fourth line is by the the independence of the query–response pairs $\{c_i, r_i\}_{i=1}^T$ and the conditional independence of $h_{t+1}$ and $c_{t+1}$ in (2), the fifth line is by the law of total probability to integrate over $Z$, the six line is by swapping the order of integral over $h_{t+1}$ and $Z$, and the seventh line is by substituting $P_t = (\mathcal{D}_t, c_{t+1})$ and the integral over $h_{t+1}$.

This completes the proof. $\square$

## C.2   Proof of Proposition 3.1

*Proof Sketch.* This proof utilizes Bayes' rule and the properties of Gaussian distributions, such as the Gaussian marginalization (see (Williams & Rasmussen, 2006, Chapter 2.1)). $\square$

*Proof.* Recall that $v_i = v^\star(r_i)$ for $i \in [T]$, then we have the following by substituting $r_i$ with $v_i$ in Lemma 2.1

$$
P(v_{t+1} \mid P_t) = \int_{\mathcal{Z}} \mathrm{d}Z \cdot P(v_{t+1} \mid P_t, Z) \cdot P(Z \mid \mathcal{D}_t).
$$

Following (Zhang et al., 2022), we derive the conditional distributions $P(v_{t+1} \mid P_t, Z)$ and $P(Z \mid \mathcal{D}_t)$ as follows.

**Deriving $P(v_{t+1} \mid P_t, Z)$.** By (11), given a hidden concept $Z \in \mathbb{R}^{d_v \times d_\phi}$, we have the formulation $v_{t+1} = Z\phi(k^\star(c_{t+1})) + \eta_{t+1}$, where $\eta_{t+1} \in N(0, \sigma_1^2 I_{d_v})$ are Gaussian noises. Recall that $q_{t+1} = k^\star(c_{t+1})$, and $v_{t+1} - Z\phi(k^\star(c_{t+1}))$ follows distribution $N(0, \sigma_1^2 I_{d_v})$, then we have

$$
P(v_{t+1} \mid P_t, Z) \propto \exp\left( -\frac{-\|v_{t+1} - Z\phi(q_{t+1})\|_2^2}{2\sigma_1^2} \right).
$$

**Deriving $P(Z \mid \mathcal{D}_t)$.** Recall that $Z$ follows Gaussian distribution $N(0, \lambda^2 I_{d_v \times d_\phi})$, $\mathcal{D}_t = (C_t, R_t)$, $K_t = k^\star(C_t)$, and $V_t = v^\star(R_t)$. By Bayes' rule, we have

$$P(Z \mid \mathcal{D}_t) = \frac{P(V_t \mid Z, \mathcal{D}_t) P(Z)}{P(V_t \mid \mathcal{D}_t)}$$

$$\propto \exp\left(-\frac{1}{2\sigma_1^2} \|V_t - Z\phi(K_t)\|_F^2\right) \cdot \exp\left(-\frac{1}{2\lambda^2} \|Z\|_F^2\right)$$

$$= \exp\left(-\frac{1}{2\sigma_1^2} \operatorname{Tr}\left[(V_t - Z\phi(K_t))(V_t - Z\phi(K_t))^\top\right] - \frac{1}{2\lambda^2} \operatorname{Tr}\left[ZZ^\top\right]\right)$$

$$= \exp\left(-\frac{1}{2\sigma_1^2} \operatorname{Tr}\left[V_t V_t^\top - 2Z\phi(K_t)V_t^\top + \left(Z\phi(K_t)\right)\left(Z\phi(K_t)\right)^\top\right] - \frac{1}{2\lambda^2} \operatorname{Tr}[ZZ^\top]\right)$$

$$= \exp\left(-\operatorname{Tr}\left[\frac{1}{2\sigma_1^2} Z\phi(K_t)\phi(K_t)^\top Z^\top + \frac{1}{2\lambda^2} ZZ^\top\right] + \operatorname{Tr}\left[\frac{1}{\sigma_1^2} Z\phi(K_t)V_t^\top\right] - \operatorname{Tr}\left[\frac{1}{2\sigma_1^2} V_t V_t^\top\right]\right)$$

$$= \exp\left(-\frac{1}{2} \operatorname{Tr}\left[Z\left(\frac{1}{\lambda^2}I + \frac{1}{\sigma_1^2}\phi(K_t)\phi(K_t)^\top\right)Z^\top\right] + \operatorname{Tr}\left[Z\left(\frac{\phi(K_t)V_t^\top}{\sigma_1^2}\right)\right] - \operatorname{Tr}\left[\frac{V_t V_t^\top}{2\sigma_1^2}\right]\right)$$

$$\propto \exp\left(-\frac{1}{2} \operatorname{Tr}\left[Z\left(\frac{1}{\lambda^2}I + \frac{1}{\sigma_1^2}\phi(K_t)\phi(K_t)^\top\right)Z^\top\right] + \operatorname{Tr}\left[Z\left(\frac{\phi(K_t)V_t^\top}{\sigma_1^2}\right)\right]\right),$$

where the first line is by Bayes' rule, the second line is by $V_t - Z\phi(K_t) \sim N(0, I_{d_v \times t})$, $Z \sim N(0, I_{d_v \times d_\phi})$, and $P(V_t \mid \mathcal{D}_t)$ is a constant value ($V_t = v^\star(R_t)$ and $\mathcal{D}_t = (C_t, R_t)$), the third line is by $\|A\|_F^2 = \operatorname{Tr}(A)$ for any matrix $A \in \mathbb{R}^{d_1 \times d_1}$ ($d_1 \in \mathbb{N}^+$), the fourth line is by $\operatorname{Tr}(A) = \operatorname{Tr}(A^\top)$ for any matrix $A \in \mathbb{R}^{d_1 \times d_1}$, the fifth and sixth lines are by rearranging terms, and the seventh line is by omitting constant terms with respect to the variable $Z$.

We define matrices

$$\Sigma_{Z,t} := \frac{1}{\lambda^2}I + \frac{1}{\sigma_1^2}\phi(K_t)\phi(K_t)^\top, \quad \text{and} \quad M := \frac{\phi(K_t)V_t^\top}{\sigma_1^2}.$$

Then we have

$$P(Z \mid \mathcal{D}_t) \propto \exp\left(-\frac{1}{2} \operatorname{Tr}\left[Z\left(\frac{1}{\lambda^2}I + \frac{1}{\sigma_1^2}\phi(K_t)\phi(K_t)^\top\right)Z^\top\right] + \operatorname{Tr}\left[Z\left(\frac{\phi(K_t)V_t^\top}{\sigma_1^2}\right)\right]\right)$$

$$= \exp\left(-\frac{1}{2} \operatorname{Tr}[Z\Sigma_{Z,t}Z^\top - 2ZM]\right)$$

$$= \exp\left(-\frac{1}{2} \operatorname{Tr}\left[\left(Z - M^\top \Sigma_{Z,t}^{-1}\right)\Sigma_{Z,t}\left(Z - M^\top \Sigma_{Z,t}^{-1}\right)^\top\right] + \frac{1}{2}\operatorname{Tr}\left[M^\top \Sigma_{Z,t}^{-1} M\right]\right)$$

$$\propto \exp\left(-\frac{1}{2} \operatorname{Tr}\left[\left(Z - M^\top \Sigma_{Z,t}^{-1}\right)\Sigma_{Z,t}\left(Z - M^\top \Sigma_{Z,t}^{-1}\right)^\top\right]\right),$$

where the second line is by the definition of $\Sigma_{Z,t}$ and $M$, the third line is by adding and subtracting terms and rearranging them, and the fourth line is by omitting constant terms with respect to the $Z - M^\top \Sigma_{Z,t}^{-1}$.

We define

$$\bar{Z}_t = M^\top \Sigma_{Z,t}^{-1}$$

$$= V_t \phi(K_t)^\top \left(\frac{\sigma_1^2}{\lambda^2}I + \phi(K_t)\phi(K_t)^\top\right)^{-1}$$

$$= V_t \left(\frac{\sigma_1^2}{\lambda^2}I + \phi(K_t)\phi(K_t)^\top\right)^{-1}\phi(K_t)^\top. \qquad \text{(By Woodbury identity)}$$

**Deriving $P(v_{t+1} \mid P_t)$.** By the distribution $P(v_{t+1} \mid P_t, Z)$ and $P(Z \mid \mathcal{D}_t)$, we have

$$P(v_{t+1} \mid P_t) = \int_{\mathcal{Z}} \mathrm{d}Z \cdot P(v_{t+1} \mid P_t, Z) \cdot P(Z \mid \mathcal{D}_t)$$

$$= \int_{\mathcal{Z}} \mathrm{d}Z \cdot N(v_{t+1}; Z\phi(q_{t+1}), \sigma_1^2 I_{d_v}) \cdot N(Z; \bar{Z}_t, \Sigma_{Z,t}^{-1}).$$

By standard results on Gaussian marginalization (Williams & Rasmussen, 2006, Chapter 2.1), $P(v_{t+1} \mid P_t)$ also follows Gaussian distribution, with mean $\mu_t$ and covariance $\text{Cov}_t$ as follows:

$$\mu_t = \mathbb{E}\left[Z\phi(q_{t+1}) + \eta_{t+1} \mid P_t\right] == \mathbb{E}\left[Z \mid P_t\right]\phi\left(q_{t+1}\right) = \bar{Z}_t\phi\left(q_{t+1}\right).$$

$$\text{Cov}_t = \text{Cov}\left[Z\phi\left(q_{t+1}\right) + \eta_{t+1} \mid P_t\right] = \phi\left(q_{t+1}\right)^\top \Sigma_{Z,t}^{-1}\phi\left(q_{t+1}\right) + \sigma_1^2 I.$$

Therefore, we obtain

$$P(v_{t+1} \mid P_t) \propto \exp\left(-\frac{\left\|v_{t+1} - \bar{Z}_t\phi(q_{t+1})\right\|_2^2}{2\Sigma_t}\right),$$

where

$$\bar{Z}_t = V_t(\frac{\sigma_1^2}{\lambda^2}I + \phi(K_t)\phi(K_t)^\top)^{-1}\phi(K_t)^\top, \ \Sigma_{Z,t} = \frac{1}{\lambda^2}I + \frac{1}{\sigma_1^2}\phi(K_t)\phi(K_t)^\top, \ \Sigma_t = \sigma_1^2 I + \phi(q_{t+1})^\top\Sigma_{Z,t}^{-1}\phi(q_{t+1}).$$

This completes the proof. $\qquad\square$

## C.3 Generalization Bound for In-Context Learning

**Proof of Theorem 3.1.** Referring to (Li et al., 2023), we view in-context learning as an algorithm learning problem. Based on this, we derive the generalization bound for in-context learning with the $\epsilon$-stability of the algorithms.

First, we give the definition of $\epsilon$-stability.

**Definition C.1** ($\epsilon$-Stability). Given a Softmax-attention model with parameter $\theta$, and some possible input prompt examples $\mathcal{D}_t \coloneqq \{c_i, r_i\}_{i=1}^t$ of a constant size $t$. For any test query $c_{t+1}$, we denote the loss function with respect to $(\mathcal{D}_t, c_{t+1})$ as $L(\mathcal{D}_t, c_{t+1})$. The in-context learning of this Softmax-attention model is $\epsilon$-uniformly stable if, for any two prompt $\mathcal{D}_t$ and $\bar{\mathcal{D}}_t$ with only one different prompt example, we have

$$\sup_{c_{t+1}} \mathbb{E}_{\mathcal{D}_t}\left[L(\mathcal{D}_t, c_{t+1}) - L(\bar{\mathcal{D}}_t, c_{t+1})\right] \leq \epsilon, \tag{15}$$

where the expectation is taken over the randomness of prompt examples $\mathcal{D}_t$.

Next, we give the $\epsilon$-stability of the one-layer attention model.

**Lemma C.1** (Lemma B.1 of (Li et al., 2023)). *Let $x, \epsilon \in \mathbb{R}^n$ be vectors satisfying $\|x\|_\infty, \|x + \epsilon\|_\infty < c$. Then, we have*

$$\|\text{Softmax}(x)\|_\infty \leq \frac{e^{2c}}{n}, \quad \text{and} \quad \|\text{Softmax}(x) - \text{Softmax}(x + \epsilon)\|_1 \leq \frac{2e^{2c}\|\epsilon\|_1}{n}.$$

*Proof of Lemma C.1.* For any $x \in \mathbb{R}^d$, we define $\text{Softmax}(x_i)$ as

$$\text{Softmax}(x_i) = \frac{e^{x_i}}{\sum_{j=1}^n e^{x_j}}, \quad i = 1, \ldots, d.$$

Then we have

$$\text{Softmax}(x) = \begin{bmatrix} \text{Softmax}(x_1) \\ \text{Softmax}(x_2) \\ \vdots \\ \text{Softmax}(x_d) \end{bmatrix} = \frac{1}{\sum_{j=1}^n e^{x_j}} \begin{bmatrix} e^{x_1} \\ e^{x_2} \\ \vdots \\ e^{x_d} \end{bmatrix} \in \mathbb{R}^d.$$

Using the monotonicity of $\mathrm{Softmax}$ function, we have

$$
\begin{aligned}
\|\mathrm{Softmax}(x)\|_\infty &= \frac{e^{\max_i x_i}}{\sum_{i=1}^n e^{x_i}} && \text{(By the definition of $\mathrm{Softmax}$)} \\
&\leq \frac{e^c}{e^c + \sum_{i=2}^n e^{x_i}} && \text{(By the monotonicity and $x_i \leq c$)} \\
&\leq \frac{e^c}{e^c + \sum_{i=2}^n e^{-c}} && \text{(By $x_j \geq -c$, $j \neq i$)} \\
&= \frac{e^{2c}}{e^{2c} + n - 1} \\
&\leq \frac{e^{2c}}{n}, && \text{(By $e^{2c} \geq 1$)}
\end{aligned}
$$

where the fourth line is by multiplying $e^c$ to both the numerator and the denominator.

By taking the derivative of $\mathrm{Softmax}(x_i)$ with respect to $x_k$, we have

$$
\frac{\partial\, \mathrm{Softmax}(x_i)}{\partial x_k} = \frac{\partial}{\partial x_k}\left(\frac{e^{x_i}}{\sum_{j=1}^n e^{x_j}}\right) = \frac{e^{x_i} \cdot \delta_{ik} \cdot \sum_j e^{x_j} - e^{x_i} \cdot e^{x_k}}{\left(\sum_{j=1}^n e^{x_j}\right)^2} = \mathrm{Softmax}(x_i)\left(\delta_{ik} - \mathrm{Softmax}(x_k)\right),
$$

where $\delta_{ik}$ is the Kronecker delta, which equals 1 if $i = k$ and 0 otherwise.

By the definition of directional derivative along with direction vector $\epsilon \in \mathbb{R}^d$, we have

$$
\lim_{\delta \to 0} \frac{\mathrm{Softmax}(x + \delta\epsilon) - \mathrm{Softmax}(x)}{\delta} = \left[\mathrm{diag}\left(\mathrm{Softmax}(x)\right) - \mathrm{Softmax}(x)\,\mathrm{Softmax}(x)^\top\right]\epsilon. \tag{16}
$$

Furthermore, we have the following norm bound

$$
\begin{aligned}
&\left\|\left[\mathrm{diag}\left(\mathrm{Softmax}(x)\right) - \mathrm{Softmax}(x)\,\mathrm{Softmax}(x)^\top\right]\epsilon\right\|_1 \\
&\leq \left\|\left[\mathrm{diag}\left(\mathrm{Softmax}(x)\right) - \mathrm{Softmax}(x)\,\mathrm{Softmax}(x)^\top\right]\right\|_{1,\infty} \cdot \|\epsilon\|_1 \\
&= \max_i\left\{\left(\frac{e^{x_i}}{\sum_{j=1}^n e^{x_j}} - \frac{e^{2x_i}}{(\sum_{j=1}^n e^{x_j})^2}\right) + \sum_{j\neq i}\frac{e^{x_i}\cdot e^{x_j}}{(\sum_{j=1}^n e^{x_j})^2}\right\} \cdot \|\epsilon\|_1 \\
&= \max_i\left\{\frac{2e^{x_i}\cdot \sum_{j\neq i} e^{x_j}}{(\sum_{j=1}^n e^{x_j})^2}\right\} \cdot \|\epsilon\|_1 \\
&\leq \frac{2e^{2c}\|\epsilon\|_1}{n},
\end{aligned}
$$

where the second line is by the norm upper bound $\|A\epsilon\|_1 \leq \|A\|_{1,\infty} \cdot \|\epsilon\|_1$ for any $A \in \mathbb{R}^{d\times d}$ ($d \in \mathbb{N}^+$), the third line is by the definition of $\ell_{1,\infty}$ norm of matrix, and the fourth line is by simple algebraic calculations.

Building on this observation, we bound the left-hand side of (16) by $2e^{2c}\|\epsilon\|_1/n$. Integrating that derivative (left side of (16)) from $\delta = 0$ to 1 then yields the desired result

$$
\|\mathrm{Softmax}(x) - \mathrm{Softmax}(x + \epsilon)\|_1 \leq \frac{2e^{2c}\|\epsilon\|_1}{n}.
$$

This completes the proof. $\qquad\qquad\square$

Then we have the following lemma to characterize the stability of the one-layer attention model under our in-context memory reshaping phenomenon.

**Lemma C.2** (Stability of One-Layer Attention, Modified from Lemma B.2 of (Li et al., 2023)). *Let $C_t = [c_1 \cdots c_t]$ and $R_t = [r_1 \cdots r_t]$. We denote $\widetilde{W}_K^\star = W_K^\star C_t(W_K^\star \widetilde{C}_t)^\dagger W_K^\star$ and $\widetilde{W}_V^\star = W_V^\star R_t(W_V^\star \widetilde{R}_t)^\dagger W_V^\star$. Let the weight matrices be bounded as $\|(\widetilde{W}_K^\star)^\top W_K^\star\|_{2,2} \leq \Gamma$ and $\|\widetilde{W}_V^\star\|_{2,2} \leq 1$. Let $E_1 = [\epsilon_{1,1} \cdots \epsilon_{1,t}]$ and $E_2 = [\epsilon_{2,1} \cdots \epsilon_{2,t}]$ be the perturbation.*

We set $\bar{C}_t = C_t + E_1$, $\bar{c}_{t+1} = c_{t+1} + \epsilon_{1,t+1}$, and $\bar{R}_t = R_t + E_2$. Assume that $c_i, r_i, c_i + \epsilon_{1,i}$, and $r_i + \epsilon_{2,i}$ lies in the unit ball, i.e., $\|C_t\|_{2,\infty}, \|R_t\|_{2,\infty}, \|C_t + E_1\|_{2,\infty}, \|R_t + E_2\|_{2,\infty}, \|c_{t+1} + \epsilon_{1,t+1}\|_2, \|r_{t+1} + \epsilon_{2,t+1}\|_2 \leq 1$. Additionally, for any $i \in [t]$, we assume $\|\epsilon_{1,i}\|_2, \|\epsilon_{2,i}\|_2 \leq C_0/t$, and $\epsilon_{1,t+1} \leq C_0/t$, where $C_0 > 0$ is an absolute constant.

We define the output of Softmax-attention for the input examples as

$$v_{t+1} = \widetilde{W}_V^\star R_t \, \mathrm{Softmax}\left((\widetilde{W}_K^\star C_t)^\top W_K^\star c_{t+1}\right), \quad and \quad \bar{v}_{t+1} = \widetilde{W}_V^\star \bar{R}_t \, \mathrm{Softmax}\left((\widetilde{W}_K^\star \bar{C}_t)^\top W_K^\star \bar{c}_{t+1}\right). \tag{17}$$

Let $\mathcal{D}_t = \{c_i, r_i\}_{i \in [t]}$, and $\bar{\mathcal{D}}_t = \{\bar{c}_i, \bar{r}_i\}_{i \in [t]}$. We further denote the true response of $c_{t+1}$ as $r_{t+1}$, and define the loss function

$$L(\mathcal{D}_t, c_{t+1}) = \mathcal{L}\left(v_{t+1}, \widetilde{W}_V^\star r_{t+1}\right), \quad and \quad L(\bar{\mathcal{D}}_t, c_{t+1}) = \mathcal{L}(\bar{v}_{t+1}, \widetilde{W}_V^\star r_{t+1}),$$

where $\mathcal{L}$ is a task-specific loss function and follows Assumption 3.3, e.g., mean squared error loss with a bounded input domain.

Then we have

$$\|\bar{v}_{t+1} - v_{t+1}\|_2 \leq \frac{C_0(4\Gamma + 1)e^{2\Gamma}}{t}, \tag{18}$$

and

$$\sup_{c_{t+1}} \mathbb{E}_{\mathcal{D}_t} \left[L(\bar{\mathcal{D}}_t, c_{t+1}) - L(\mathcal{D}_t, c_{t+1})\right] \leq \frac{MC_0(4\Gamma + 1)e^{2\Gamma}}{t}. \tag{19}$$

Specifically, $\epsilon$-stability (Definition C.1) corresponds to perturbing only one prompt example. As a result, we obtain $\epsilon = MC_0(4\Gamma + 1)e^{2\Gamma}/t$.

*Proof of Lemma C.2.* Consider the output difference $\Delta := \bar{v}_{t+1} - v_{t+1}$. By (17), we have

$$\begin{aligned}
\Delta &= \widetilde{W}_V^\star \bar{R}_t \, \mathrm{Softmax}\left((\widetilde{W}_K^\star \bar{C}_t)^\top W_K^\star \bar{c}_{t+1}\right) - \widetilde{W}_V^\star R_t \, \mathrm{Softmax}\left((\widetilde{W}_K^\star C_t)^\top W_K^\star c_{t+1}\right) \\
&= \widetilde{W}_V^\star (R_t + E_2) \, \mathrm{Softmax}\left((\widetilde{W}_K^\star \bar{C}_t)^\top W_K^\star \bar{c}_{t+1}\right) - \widetilde{W}_V^\star R_t \, \mathrm{Softmax}\left((\widetilde{W}_K^\star C_t)^\top W_K^\star c_{t+1}\right) \\
&= \underbrace{\widetilde{W}_V^\star R_t \left[\mathrm{Softmax}\left(\bar{C}_t^\top (\widetilde{W}_K^\star)^\top W_K^\star \bar{c}_{t+1}\right) - \mathrm{Softmax}\left(C_t^\top (\widetilde{W}_K^\star)^\top W_K^\star c_{t+1}\right)\right]}_{\Delta_1} \\
&\quad + \underbrace{\widetilde{W}_V^\star E_2 \, \mathrm{Softmax}\left(\bar{C}_t^\top (\widetilde{W}_K^\star)^\top W_K^\star \bar{c}_{t+1}\right)}_{\Delta_2},
\end{aligned}$$

where the second line is by $\bar{R}_t = R_t + E_2$, and the third line is by rearranging the terms.

To bound $\Delta_1$ and $\Delta_2$, we first establish the following norm estimates.

Recall that $\|(\widetilde{W}_K^\star)^\top W_K^\star\|_{2,2} \leq \Gamma$, $\|\bar{C}_t\|_{2,\infty} = \|C_t + E_1\|_{2,\infty} \leq 1$, and $\|c_{t+1}\|_2, \|\bar{c}_{t+1}\|_2 \leq 1$, then we have

$$\left\|\bar{C}_t^\top (\widetilde{W}_K^\star)^\top W_K^\star \bar{c}_{t+1}\right\|_\infty \leq \|\bar{C}_t\|_{2,\infty} \cdot \left\|(\widetilde{W}_K^\star)^\top W_K^\star\right\|_{2,2} \cdot \|\bar{c}_{t+1}\|_2 \leq \Gamma, \tag{20}$$

where the inequality is by $\|ABx\|_\infty \leq \|A^\top\|_{2,\infty} \|B\|_{2,2} \|x\|_2$ for any $A \in \mathbb{R}^{d_1 \times d_2}$, $B \in \mathbb{R}^{d_2 \times d_3}$, and $x \in \mathbb{R}^{d_3}$ $(d_1, d_2, d_3 \in \mathbb{N}^+)$.

Similarly, with $\|c_{t+1}\|_2 \leq 1$, we have

$$\left\|\bar{C}_t^\top (\widetilde{W}_K^\star)^\top W_K^\star c_{t+1}\right\|_\infty \leq \|\bar{C}_t\|_{2,\infty} \cdot \left\|(\widetilde{W}_K^\star)^\top W_K^\star\right\|_{2,2} \cdot \|c_{t+1}\|_2 \leq \Gamma, \tag{21}$$

where the inequality is by $||ABx||_\infty \leq ||A^\top||_{2,\infty}||B||_{2,2}||x||_2$ for any $A \in \mathbb{R}^{d_1 \times d_2}$, $B \in \mathbb{R}^{d_2 \times d_3}$, and $x \in \mathbb{R}^{d_3}$ ($d_1, d_2, d_3 \in \mathbb{N}^+$).

By $||C_t||_{2,\infty} \leq 1$, we have

$$\left\| C_t^\top (\widetilde{W}_K^\star)^\top W_K^\star c_{t+1} \right\|_\infty \leq ||C_t||_{2,\infty} \cdot \left\| (\widetilde{W}_K^\star)^\top W_K^\star \right\|_{2,2} \cdot ||c_{t+1}||_2 \leq \Gamma, \tag{22}$$

where the inequality is by $||ABx||_\infty \leq ||A^\top||_{2,\infty}||B||_{2,2}||x||_2$ for any $A \in \mathbb{R}^{d_1 \times d_2}$, $B \in \mathbb{R}^{d_2 \times d_3}$, and $x \in \mathbb{R}^{d_3}$ ($d_1, d_2, d_3 \in \mathbb{N}^+$).

We further have

$$\left\| \bar{C}_t^\top (\widetilde{W}_K^\star)^\top W_K^\star \epsilon_{1,t+1} \right\|_1 \leq \left\| \bar{C}_t \right\|_{2,1} \cdot \left\| (\widetilde{W}_K^\star)^\top W_K^\star \right\|_{2,2} \cdot ||\epsilon_{1,t+1}||_2$$

$$= \sum_{i=1}^t ||\bar{c}_i||_2 \cdot \left\| (\widetilde{W}_K^\star)^\top W_K^\star \right\|_{2,2} \cdot ||\epsilon_{1,t+1}||_2$$

$$\leq C_0 \Gamma, \tag{23}$$

where the first line is by $||ABx||_1 \leq ||A^\top||_{2,1}||B||_{2,2}||x||_2$ for any $A \in \mathbb{R}^{d_1 \times d_2}$, $B \in \mathbb{R}^{d_2 \times d_3}$, and $x \in \mathbb{R}^{d_3}$ ($d_1, d_2, d_3 \in \mathbb{N}^+$), the second line is by the definition of $\ell_{2,1}$ norm of a matrix, and the third line is by $||\bar{c}_i||_2 \leq 1$ and $||\epsilon_{1,t+1}||_2 \leq C_0/t$.

Similarly, we have

$$\left\| E_1^\top (\widetilde{W}_K^\star)^\top W_K^\star c_{t+1} \right\|_1 \leq ||E_1||_{2,1} \cdot \left\| (\widetilde{W}_K^\star)^\top W_K^\star \right\|_{2,2} \cdot ||c_{t+1}||_2$$

$$= \sum_{i=1}^t ||\epsilon_{1,i}||_2 \cdot \left\| (\widetilde{W}_K^\star)^\top W_K^\star \right\|_{2,2} \cdot ||c_{t+1}||_2$$

$$\leq C_0 \Gamma, \tag{24}$$

where the first line is by $||ABx||_1 \leq ||A^\top||_{2,1}||B||_{2,2}||x||_2$ for any $A \in \mathbb{R}^{d_1 \times d_2}$, $B \in \mathbb{R}^{d_2 \times d_3}$, and $x \in \mathbb{R}^{d_3}$ ($d_1, d_2, d_3 \in \mathbb{N}^+$), the second line is by the definition of $\ell_{2,1}$ norm of a matrix, and the third line is by $||\epsilon_{1,i}||_2 \leq C_0/t$ and $||c_{t+1}||_2 \leq 1$.

Then we bound $\Delta_1$ and $\Delta_2$ using Lemma C.1 together with the above norm bounds.

**Bounding $\Delta_1$.** For $\Delta_1$, we have

$$\|\Delta_1\|_2 = \left\| \widetilde{W}_V^\star R_t \left[ \text{Softmax}\left( \bar{C}_t^\top (\widetilde{W}_K^\star)^\top W_K^\star \bar{c}_{t+1} \right) - \text{Softmax}\left( C_t^\top (\widetilde{W}_K^\star)^\top W_K^\star c_{t+1} \right) \right] \right\|_2$$

$$\leq \left\| \widetilde{W}_V^\star R_t \right\|_{2,\infty} \left\| \text{Softmax}\left( \bar{C}_t^\top (\widetilde{W}_K^\star)^\top W_K^\star \bar{c}_{t+1} \right) - \text{Softmax}\left( C_t^\top (\widetilde{W}_K^\star)^\top W_K^\star c_{t+1} \right) \right\|_1$$

$$\leq \left\| \text{Softmax}\left( \bar{C}_t^\top (\widetilde{W}_K^\star)^\top W_K^\star \bar{c}_{t+1} \right) - \text{Softmax}\left( C_t^\top (\widetilde{W}_K^\star)^\top W_K^\star c_{t+1} \right) \right\|_1$$

$$\leq \left\| \text{Softmax}\left( \bar{C}_t^\top (\widetilde{W}_K^\star)^\top W_K^\star \bar{c}_{t+1} \right) - \text{Softmax}\left( \bar{C}_t^\top (\widetilde{W}_K^\star)^\top W_K^\star c_{t+1} \right) \right\|_1$$

$$+ \left\| \text{Softmax}\left( \bar{C}_t^\top (\widetilde{W}_K^\star)^\top W_K^\star c_{t+1} \right) - \text{Softmax}\left( C_t^\top (\widetilde{W}_K^\star)^\top W_K^\star c_{t+1} \right) \right\|_1,$$

where the second line is by $||Ax||_2 \leq ||A||_{2,\infty}||x||_1$ for any $A \in \mathbb{R}^{d_1 \times d_2}$, and $x \in \mathbb{R}^{d_2}$ ($d_1, d_2 \in \mathbb{N}^+$), the third line is by $||\widetilde{W}_V^\star||_{2,2}, ||R_t||_{2,\infty} \leq 1$, and $||\widetilde{W}_V^\star R_t||_{2,\infty} \leq ||\widetilde{W}_V^\star||_{2,2} \cdot ||R_t||_{2,\infty} \leq 1$, and the fourth line is by triangle inequality.

Then we bound the above two terms. For the first term, we have:

$$\left\| \text{Softmax}\left( \bar{C}_t^\top (\widetilde{W}_K^\star)^\top W_K^\star \bar{c}_{t+1} \right) - \text{Softmax}\left( \bar{C}_t^\top (\widetilde{W}_K^\star)^\top W_K^\star c_{t+1} \right) \right\|_1$$

$$= \left\| \text{Softmax}\left( \bar{C}_t^\top (\widetilde{W}_K^\star)^\top W_K^\star (c_{t+1} + \epsilon_{1,t+1}) \right) - \text{Softmax}\left( \bar{C}_t^\top (\widetilde{W}_K^\star)^\top W_K^\star c_{t+1} \right) \right\|_1$$

$$\leq \frac{2e^{2\Gamma}}{t} \left\| \bar{C}_t^\top (\widetilde{W}_K^\star)^\top W_K^\star \epsilon_{1,t+1} \right\|_1 \qquad \text{(By Lemma C.1, (20), and (21))}$$

$$\leq \frac{2C_0 \Gamma e^{2\Gamma}}{t}. \qquad \text{(By (23))}$$

For the second term, we have:

$$\left\| \text{Softmax}\left(\bar{C}_t^\top (\widetilde{W}_K^\star)^\top W_K^\star c_{t+1}\right) - \text{Softmax}\left(C_t^\top (\widetilde{W}_K^\star)^\top W_K^\star c_{t+1}\right) \right\|_1$$

$$= \left\| \text{Softmax}\left((C_t + E_1)^\top (\widetilde{W}_K^\star)^\top W_K^\star c_{t+1}\right) - \text{Softmax}\left(C_t^\top (\widetilde{W}_K^\star)^\top W_K^\star c_{t+1}\right) \right\|_1$$

$$\leq \frac{2e^{2\Gamma}}{t} \left\| E_1^\top (\widetilde{W}_K^\star)^\top W_K^\star c_{t+1} \right\|_1 \qquad \text{(By Lemma C.1, (21), and (22))}$$

$$\leq \frac{2C_0 \Gamma e^{2\Gamma}}{t}. \qquad \text{(By (24))}$$

Combining the above two terms, we have

$$\|\Delta_1\|_2 \leq \frac{4C_0 \Gamma e^{2\Gamma}}{t}.$$

**Bounding $\Delta_2$.** For $\Delta_2$, we have

$$\|\Delta_2\|_2 = \left\| \widetilde{W}_V^\star E_2 \, \text{Softmax}\left(\bar{C}_t^\top (\widetilde{W}_K^\star)^\top W_K^\star \bar{c}_{t+1}\right) \right\|_2$$

$$\leq \left\| \widetilde{W}_V^\star \right\|_{2,2} \cdot \|E_2\|_{2,1} \cdot \left\| \text{Softmax}\left(\bar{C}_t^\top (\widetilde{W}_K^\star)^\top W_K^\star \bar{c}_{t+1}\right) \right\|_\infty$$

$$\leq \left\| \widetilde{W}_V^\star \right\|_{2,2} \cdot \sum_{i=1}^t \|\epsilon_{2,i}\|_2 \cdot \left\| \text{Softmax}\left(\bar{C}_t^\top (\widetilde{W}_K^\star)^\top W_K^\star \bar{c}_{t+1}\right) \right\|_\infty$$

$$\leq C_0 \frac{e^{2\Gamma}}{t},$$

where the second line is by $\|ABx\|_2 \leq \|A\|_{2,2}\|B\|_{2,1}\|x\|_2$ for any $A \in \mathbb{R}^{d_1 \times d_2}$, $B \in \mathbb{R}^{d_2 \times d_3}$, and $x \in \mathbb{R}^{d_3}$ ($d_1, d_2, d_3 \in \mathbb{N}^+$), the third line is by the definition of the $\ell_{2,1}$ norm of a matrix, and the fourth line is by $\|\widetilde{W}_V^\star\|_{2,2} \leq 1$, $\|\epsilon_{2,i}\| \leq C_0/t$, Lemma C.1 and (20).

Combining the above results, we get the desired result:

$$\Delta \leq \frac{C_0(4\Gamma + 1)e^{2\Gamma}}{t}. \tag{25}$$

Recall that in general case, $L(\mathcal{D}_t, c_{t+1}) = \mathcal{L}(v_{t+1}, \widetilde{W}_V^\star r_{t+1})$, and $L(\bar{\mathcal{D}}_t, c_{t+1}) = \mathcal{L}(\bar{v}_{t+1}, \widetilde{W}_V^\star r_{t+1})$ where $\mathcal{L}$ is a task-specific loss function. Then for the difference of the loss function, we have

$$\sup_{c_{t+1}} \mathbb{E}_{\mathcal{D}_t}\left[ L(\bar{\mathcal{D}}_t, c_{t+1}) - L(\mathcal{D}_t, c_{t+1}) \right] = \sup_{c_{t+1}} \mathbb{E}_{\mathcal{D}_t}\left[ \mathcal{L}(\bar{v}_{t+1}, \widetilde{W}_V^\star r_{t+1}) - \mathcal{L}\left(v_{t+1}, \widetilde{W}_V^\star r_{t+1}\right) \right]$$

$$\leq \sup_{c_{t+1}} \mathbb{E}_{\mathcal{D}_t}\left[ M\|\bar{v}_{t+1} - v_{t+1}\|_2 \right] \qquad \text{(By Assumption 3.3)}$$

$$\leq \frac{MC_0(4\Gamma + 1)e^{2\Gamma}}{t}. \qquad \text{(By (25))}$$

This completes the proof. $\qquad \square$

Now, we introduce one lemma about the relation between $\epsilon$-stability and generalization bound, based on the Lemma 7 in (Bousquet & Elisseeff, 2002). Here we transfer the description in the original lemma to the in-context learning setting.

**Lemma C.3** (Connection between Generalization Bound and $\epsilon$-Stability, Lemma 7 of (Bousquet & Elisseeff, 2002)). *We denote the generalization error of in-context learning with prompt $P_t$ and test query $c_{t+1}$ as $R(\mathcal{D}_t) = \mathbb{E}_{c_{t+1}}[L(\mathcal{D}_t, c_{t+1})]$. The empirical error as $R_{emp}(\mathcal{D}_t) = \frac{1}{t}\sum_{i=1}^t L(\mathcal{D}_t, c_i)$, i.e., the average loss value when using $c_i$ as the test query respectively. Then the generalization bound is as follows:*

$$\mathbb{E}_{\mathcal{D}_t}\left[ R(\mathcal{D}_t) - R_{emp}(\mathcal{D}_t) \right] = \mathbb{E}_{\mathcal{D}_t, (\bar{c}_i, \bar{r}_i)}\left[ L(\mathcal{D}_t, \bar{c}_i) - L(\bar{\mathcal{D}}_t^i, \bar{c}_i) \right], \tag{26}$$

*where $\bar{\mathcal{D}}_t^i$ denotes replace $(c_i, r_i)$ with $(\bar{c}_i, \bar{r}_i)$ in $\mathcal{D}_t$.*

We give the following proof of Theorem 3.1 based on the above results.

*Proof of Theorem 3.1.* Using Lemma C.3, we compare the right-hand side of (26) with left-hand side of (19) in Lemma C.2, and thereby obtain the following generalization bound for in-context learning

$$\mathbb{E}_{\mathcal{D}_t}\left[R(\mathcal{D}_t) - R_{\text{emp}}(\mathcal{D}_t)\right] \leq \frac{MC_0(4\Gamma + 1)e^{2\Gamma}}{t}.$$

This completes the proof. □

# D    Experimental Details

Here we present our experimental details.

## D.1    Pre-training GPT-2 for Linear Regression

In this part, we show the details about how to pre-train a GPT-2 model for a linear regression problem. Following the pre-training method in (Garg et al., 2022), we use the batch size as 64. To construct one sample in a batch, we use the following steps: (i). Sample linear regression coefficient $\beta_i \in \mathbb{R}^{20}$ from $N(0, I)$. (ii). Generate queries $x_{i,j}$ from the Gaussian mixture model $\omega_1 N(-2, I) + \omega_2 N(2, I)$, where $\omega_1 = 1, \omega_2 = 0$ in the pre-training. Then we formalize $\{x_{i,j}\}_{j=1}^{k}$, where $k = 50$. (iii). For each query $x_{i,j}$, use $y_{i,j} = \beta_i^T x_{i,j}$ to calculate the true response. Now we generate one training sample, the model input is $[x_{i,1}, y_{i,1}, \cdots, x_{i,49}, y_{i,49}, x_{i,50}]$, and the training target is $o_i = [y_{i,1}, \cdots, y_{i,49}, y_{i,50}]$. We use the MSE loss between prediction and true value of $o_i$. The pre-training process iterates for 500k steps. We implement experiments on 1 NVIDIA A100 80GB GPU.

## D.2    Sensitivity to Covariance Shifts

In this part, we show the details about how to evaluate the model performance in the testing process. We generate samples similar to the pre-training process. The batch size is 64, and the number of batch is 100, i.e., we have 6400 samples totally. For each sample, we extend the prompt length from 49 to 74, to learn the performance of in-context learning when the prompt length is longer than we use in pre-training. The input to the model is $[x_{i,1}, y_{i,1}, \cdots, x_{i,74}, y_{i,74}, x_{i,75}]$, and the target is $o_i = [y_{i,1}, \cdots, y_{i,74}, y_{i,75}]$. For each in-context length $j \in [75]$, we calculate the R-squared between the estimation and true value for all 6400 samples.

## D.3    Sensitivity to Response's Accuracy

In this part, we show the details about how to generate samples with different accuracy of the response. (i). Firstly, we generate $i^{th}$ sample as the pre-training process in Appendix D.1, where queries $x_{i,j}$ are from the Gaussian mixture model $\omega_1 N(-2, I) + \omega_2 N(2, I), \omega_1 = 1, \omega_2 = 0$, and the prompt length becomes 75. The input is $[x_{i,1}, y_{i,1}, \cdots, x_{i,74}, y_{i,74}, x_{i,75}]$, and the target is $o_i = [y_{i,1}, \cdots, y_{i,74}, y_{i,75}]$. (ii). Next, we denote the response's accuracy as $\alpha$ (e.g., 80%). To predict $y_{i,j}$ in $i^{th}$ sample with $j - 1$ in-context prompts, we randomly permute the prompt labels $[y_{i,1}, \cdots, y_{i,j-1}]$ in the input, with a permutation rate as $1 - \alpha$. In this way, we generate samples with the accuracy of response as $\alpha$.

## D.4    Sensitivity to Similarity between Prompts and *Test Query*

In this part, we show the details about how to generate prompts with different similarities as the *test query*. (i). Firstly, we generate $i^{th}$ sample as the pre-training process in Appendix D.1, where queries $x_{i,j}$ are from the Gaussian mixture model $\omega_1 N(-2, I) + \omega_2 N(2, I), \omega_1 = 1, \omega_2 = 0$. Here we extend the prompt length from 49 to 1000. The target is still $o_i = [y_{i,1}, \cdots, y_{i,74}, y_{i,75}]$, but the input changes according the level of similarity. (ii). Secondly, we construct inputs according to the level of similarity. We consider three kinds of similarity levels: high, normal, and low similarity. For each $j \in [75]$ and $j > 1$, in the high similarity case, we select the most similar $j - 1$ prompts as $x_{i,j}$ from $\{(x_{i,k}, y_{i,k})\}_{k=1}^{1000} \backslash \{(x_{i,j}, y_{i,j})\}$. The selected $j - 1$ prompts act as the model input to predict the $y_{i,j}$. Similarly, in normal similarity case, we just use $\{(x_{i,k}, y_{i,k})\}_{k=1}^{j-1}$ as input. In low similarity case, we select the least similar $j - 1$ prompts as $x_{i,j}$ from $\{(x_{i,k}, y_{i,k})\}_{k=1}^{1000} \backslash \{(x_{i,j}, y_{i,j})\}$ to act as input. In this way, we generate prompts with different similarities as the *test query*.

## D.5    Decision Tree and 2-Layer Neural Network

Following a similar setting as previous linear regression and (Garg et al., 2022), we show the following details about the decision tree and 2-layer neural network. In addition to the results shown in Figure 2 and Figure 3, we report exact values at specific points in Table 5, Table 6, and Table 7.

For the decision tree, we consider the function $f$ as a decision tree with 20-dimensional inputs and a depth of 4. Each function $f$ uses a full binary tree with 16 leaf nodes. Non-leaf nodes specify input coordinates. Leaf nodes assign output values. To compute $f(x)$, begin at the root. Navigate through the tree. Move right if the coordinate value is positive. Move left otherwise. The output $f(x)$ comes from the leaf node reached. We choose random coordinates for non-leaf nodes from $\{1, 2, \ldots, 20\}$. Draw leaf values from a normal distribution $N(0, 1)$.

For the 2-layer neural network, we consider ReLU neural networks. We set each function $f$ as $f(x) = \sum_{i=1}^{r} \alpha_i \sigma(\mathbf{w}_i^\top x)$, where $\alpha_i \in \mathbb{R}$, $\mathbf{w}_i \in \mathbb{R}^d$, and $\sigma(\cdot) = \max(0, \cdot)$ is the ReLU activation function. We draw network parameters $\alpha_i$ and $\mathbf{w}_i$ from $N(0, 2/r)$ and $N(0, I_d)$. We use the number of hidden nodes $r$ as 100.

*Table 5.* **Sensitivity to Covariance Shifts.** We select specific values for ICL prompt number at $15, 30, 45, 60$ from Figure 2 and Figure 3. Note that performance decreases when the prompt length exceeds the pre-training length (i.e., 50), a well-known issue (Dai et al., 2019; Anil et al., 2022). We believe this is due to the absolute positional encodings in GPT-2, as noted in (Zhang et al., 2024)

| Test Distribution | Decision Tree | | | | 2-Layer Neural Network | | | |
|---|---|---|---|---|---|---|---|---|
| | 15 | 30 | 45 | 60 | 15 | 30 | 45 | 60 |
| Baseline | 0.806 | 0.862 | 0.854 | 0.875 | 0.886 | 0.946 | 0.953 | 0.962 |
| $N(-2, I)$ | 0.866 | 0.943 | **0.972** | 0.943 | 0.901 | 0.968 | **0.979** | 0.956 |
| $0.9N(-2, I) + 0.1N(2, I)$ | 0.863 | 0.874 | **0.967** | 0.943 | 0.902 | 0.954 | **0.968** | 0.952 |
| $0.7N(-2, I) + 0.3N(2, I)$ | 0.408 | 0.658 | **0.671** | 0.654 | 0.765 | 0.845 | **0.881** | 0.858 |

*Table 6.* **Sensitivity to Response's Accuracy.** We select specific values for in-context learning prompt number at $15, 30, 45, 60$ from Figure 2 and Figure 3.

| Response Accuracy | Decision Tree | | | | 2-Layer Neural Network | | | |
|---|---|---|---|---|---|---|---|---|
| | 15 | 30 | 45 | 60 | 15 | 30 | 45 | 60 |
| Baseline | 0.505 | 0.801 | 0.733 | 0.849 | 0.907 | 0.924 | 0.953 | 0.961 |
| 100% | 0.737 | 0.964 | **0.967** | 0.858 | 0.921 | 0.960 | **0.978** | 0.964 |
| 90% | 0.737 | 0.921 | **0.967** | 0.875 | 0.921 | 0.947 | **0.964** | 0.959 |
| 80% | 0.737 | 0.921 | **0.948** | 0.796 | 0.902 | 0.903 | **0.946** | 0.940 |

## D.6  GPT-J on Sentiment Classification Task

In this part, we evaluate the in-context learning (ICL) behavior of the GPT-J model, similar to (Min et al., 2022). We use the sentiment classification task with the "TweetEval: Hate Speech Detection" dataset (Basile et al., 2019).

**Sensitivity to Covariance Shifts.**   To assess how sensitive the model is to domain shifts in the in-context examples, we conduct experiments using two settings: (i) in-distribution in-context examples from the TweetEval dataset, and (ii) out-of-distribution (OOD) in-context examples from the CC-News corpus (Nagel, 2016), which consists of long-form, formal news articles. To ensure fair results, we control for the average sentence length between the two sources during sampling. We use GPT-J (6B parameters) following the protocol in (Min et al., 2022), and compare the performance across two configurations: zero-shot ($k = 0$) and few-shot ($k = 4$) with both in-distribution and OOD examples. We show the results Table 8.

*Table 7.* **Sensitivity to Similarity with *Test Query*.** We select specific values for in-context learning prompt number at $15, 30, 45, 60$ from Figure 2 and Figure 3.

| Similarity Degree | Decision Tree | | | | 2-Layer Neural Network | | | |
|---|---|---|---|---|---|---|---|---|
| | 15 | 30 | 45 | 60 | 15 | 30 | 45 | 60 |
| Baseline | 0.880 | 0.880 | 0.871 | 0.895 | 0.917 | 0.943 | 0.960 | 0.962 |
| *Best* | 0.955 | 0.976 | **0.987** | 0.971 | 0.976 | 0.988 | **0.991** | 0.988 |
| *Normal* | 0.971 | 0.957 | **0.976** | 0.942 | 0.931 | 0.972 | **0.981** | 0.967 |
| *Worst* | 0.907 | 0.937 | **0.943** | 0.927 | 0.742 | 0.866 | **0.895** | 0.882 |

*Table 8.* **Sensitivity to Covariance Shifts on Real Dataset.** Macro-F1 scores for sentiment classification ("TweetEval: Hate Speech Detection" dataset (Basile et al., 2019)) with $k = 0, 4$ in-context examples under in-distribution (No-shift) and out-of-distribution (Shift) settings.

| In-context Examples | 0 | 4 |
|---|---|---|
| No-shift | **0.3722** | **0.5313** |
| Shift | 0.3521 | 0.3830 |

**Sensitivity to Response's Accuracy.** Beyond domain alignment, we further analyze the impact of in-context response correctness on model performance. We use 4 in-context examples, and create variants of the in-context examples in which the percentage of correct query-response pairs is from $100\%$ to $25\%$. We show the results in Table 9.

*Table 9.* **Sensitivity to Response's Accuracy on Real Dataset.** Macro-F1 scores for sentiment classification ("TweetEval: Hate Speech Detection" dataset (Basile et al., 2019)) with $k = 4$ in-context examples under different proportions of correct labels.

| Response's Accuracy (%) | 100% | 75% | 50% | 25% |
|---|---|---|---|---|
| 4 In-context Examples | **0.5313** | 0.4212 | 0.4035 | 0.3640 |

### D.7 Ghost Example Construction

In this part, we show the details about how to generate the required ghost examples. Firstly, we freeze all the model parameters. For each $t \in [75]$, we generate samples as the following: We randomly choose 100 samples. For the $i$-th sample, we randomly select $x_{i,t+1}$ from $\omega_1 N(-2, I) + \omega_2 N(2, I)$, where $\omega_1 = 1, \omega_2 = 0$ or $\omega_1 = 0.5, \omega_2 = 0.5$. Then we set $\{\widetilde{c}_{i,j}, \widetilde{r}_{i,j}\}_{j \in [t]}$ as the trainable parameters. We use MSE loss to train the parameters $\{\widetilde{c}_{i,j}, \widetilde{r}_{i,j}\}_{j \in [t]}$ in each batch, until the loss convergences for this batch. We calculate the convergence loss value for one batch and use the batch loss as the ghost example construction loss for the index $t$. In this way, we get the ghost example construction loss for each $t \in [75]$ and two distribution settings, where one is the same as the training data distribution and another is different from it. The following Table 10 show the mean squared error of the ghost example construction for $t \in [75]$.

*Table 10.* **Mean Squared Error of Ghost Example Construction.** The data are from two different distributions. $15, 30, 45, 60, 75$ denote the in-context examples size.

| Data Distribution | 15 | 30 | 45 | 60 | 75 |
|---|---|---|---|---|---|
| $w_1 = 1, w_2 = 0$ | 1.50e-06 | 1.17e-05 | 3.28e-05 | 0.006 | 0.0001 |
| $w_1 = 0.5, w_2 = 0.5$ | 0.0010 | 0.0070 | 0.0049 | 0.0001 | 0.0011 |

