# OpenReview forum: "In-Context Learning as Conditioned Associative Memory Retrieval"
_ICML.cc/2025/Conference — ICML 2025 poster_

### Official Review · Reviewer_HbbS · 2025-02-22

**Overall Recommendation:** 4

**Summary:**

This paper proposes a novel interpretation of in-context learning(ICL)as the memory retrieval of the modern Hopfield model from a conditional memory set. The paper is well-written and the techniques proposed are novel.

Strengths: The innovative Hopfield-based perspective, solid theoretical framework, clear structure, and intuitive experimental focus are promising foundations for a strong paper.

Limitation: Clarify key concepts, tighten mathematical rigor, expand experiments, complete missing sections, and better highlight novelty.

**Claims And Evidence:**

The claims made in the submission are generally clear, but there is one minor problem.
Claim 2 (Hidden Concept Learning via Reshaping):
Problematic: The evidence is theoretical and relies heavily on a simplified Gaussian linear model, with no robust empirical validation in complex LLM settings. The ghost example construction (Appendix E.5) is a weak link—its small MSE values suggest feasibility, but there’s no analysis of its representativeness or stability, and the process is under-specified (e.g., optimization details). This claim needs stronger empirical support beyond linear regression.
Claim 3 (Generalization Bound) :
Problematic: The experimental support is solid for linear regression but doesn’t fully convince for broader ICL applications due to task simplicity.

**Essential References Not Discussed:**

No

**Experimental Designs Or Analyses:**

1. Sensitivity to Covariance Shifts (Section 4.1.1, Table 1, Appendix E.2)
Strengths: The design controls the test distribution shift, aligning with the claim that ICL is sensitive to covariance changes. Using multiple prompt sizes tests the generalization bound (Claim 3). Synthetic data allows precise manipulation, which is sensible for isolating this effect.
Issues:
Overly Simple Task: Linear regression is too trivial for GPT-2, which is designed for complex NLP tasks. Covariance shifts in this context (Gaussian mixtures) don’t mirror real-world ICL challenges (e.g., domain shifts in text).
Sample Size: 6400 samples (64 batches × 100) is reasonable but lacks justification for statistical power. No confidence intervals or significance tests are reported, weakening validity.
Prompt Length Extension: Extending prompts beyond pretraining (49 to 75) tests generalization but assumes GPT-2 can extrapolate, which isn’t validated.
2. Sensitivity to Response Accuracy (Section 4.1.2, Table 2, Appendix E.3)
Strengths: Permuting labels to simulate inaccuracy is a reasonable way to test sensitivity, directly addressing Claim 4. Multiple prompt sizes align with the generalization bound hypothesis.
Issues:
Permutation Method: Randomly permuting labels introduces noise but doesn’t model realistic errors (e.g., systematic biases in prompts). The permutation rate ($1 - \alpha$) is applied uniformly, which may not reflect varying error distributions.
Task Simplicity: As with covariance shifts, linear regression limits the experiment’s relevance to ICL’s typical complexity.

**Methods And Evaluation Criteria:**

1. The methods are heavily theoretical and tailored to a one-layer attention model, which oversimplifies multi-layer LLMs like GPT-2. Extending to multi-layer settings would better match the application. Ghost example construction lacks a clear benchmark or prior method for comparison, making its novelty sensible but its practicality uncertain.
2. Linear regression doesn’t capture ICL’s complexity (e.g., semantic understanding, multi-task learning), undermining applicability to real LLMs. While controlled, it lacks the richness of benchmark datasets (e.g., GLUE, SuperGLUE), reducing ecological validity.

**Other Comments Or Suggestions:**

No.

**Other Strengths And Weaknesses:**

Notation Inconsistencies: The notation is occasionally sloppy or inconsistent. For instance, on Page 2, $P_\theta(\cdot \mid C_t, R_t)$ is used to denote the LLM's predictive distribution, but $C_t$ and $R_t$ are matrices, while the conditioning typically expects a sequence.
Pseudo-Inverse Usage: In Section 3.1, the use of pseudo-inverses (e.g., $\tilde{K}_t^+$ and $\tilde{V}_t^+$) to define $W_1$ and $W_2$ assumes invertibility conditions that are not explicitly justified. If $\tilde{K}_t$ or $\tilde{V}_t$ are rank-deficient (common in high-dimensional settings), this formulation may be unstable or undefined without regularization.
Generalization Bound: The bound in Theorem 3.2, $\mathbb{E}{P_t}[R(P_t) - R{\text{emp}}(P_t)] \leq \frac{M C_0 (4\Gamma + 1) e^{2\Gamma}}{t}$, decreases as $\mathcal{O}(1/t)$, but the proof sketch relies on $\epsilon$-stability without justifying why the attention mechanism satisfies this property under the reshaping framework. The constant terms ($M$, $C_0$, $\Gamma$) are also not well-constrained, reducing practical interpretability.

**Questions For Authors:**

No.

**Relation To Broader Scientific Literature:**

The paper outlines four main contributions:
Equivalent Memory Reshaping in ICL: Prompt examples reshape pretrained weights in the attention mechanism, interpreted as memory retrieval in a modern Hopfield model (MHM).
BMA with Reshaped Memory: Constructs an explicit hidden concept expression via reshaped weights, showing ICL performs Bayesian Model Averaging (BMA) based on this concept.
Generalization Bound in ICL: Derives a bound for a one-layer attention model, showing performance improves with more prompts at a rate of $\mathcal{O}(1/t)$.
Experimental Results: Uses reshaping to explain ICL’s sensitivity to covariance shifts, response accuracy, and prompt-test similarity, validated with linear regression experiments.

**Theoretical Claims:**

The theoretical claims made in the paper proved to be correct.

---

> ### Author Rebuttal · Authors · 2025-04-01
>
> Thanks for your detailed review. We have revised our draft and addressed all concerns. **The revised version (changes marked in BLUE) and the code for external experiments are available in this [anonymous Dropbox folder](https://www.dropbox.com/scl/fo/9m982gnk45wc1w705wjsa/AKYGxykAcCFGADGQEHCN218?rlkey=h03t7ipyv8qbt9bvesfivxcsw&st=ze6gjwhu&dl=0).**
>
> > **C1**: Linear Regression
>
> **Response**
>
> Thanks for your comment. In the revised version, we extend our experimental evaluation to cover more complex tasks. Specifically, we include new results using GPT-2 on decision tree and 2-layer neural network tasks, following a similar setup as in the linear regression experiments. Additionally, we incorporate experiments with the GPT-J model on the real-world “TweetEval: Hate Speech Detection” dataset to evaluate performance on a practical sentiment classification task. Please refer to Section 4.1.4 for details.
>
> > **C2**: Extending to multi-layer setting
>
> **Response**
>
> Thanks for your insightful comment. We provide an overview of how to extend our analysis to multi-layer models. The detailed discussion appears in the Section A.2 of the revised version, and we restate it here.
>
> We regard the multi-layer transformer as iteratively retrieving memory patterns in the modern Hopfield model (MHM) through the following four steps:
>
> * Design energy function for multi-layer MHM: We design an energy function matching the multi-layer transformer [Hoover23]. The memory pattern needs retrieval after reshaping and iteration. This step is challenging.
>
> * Give clear expression of memory reshaping on model weights: This step is feasible following our framework in Sec. 3.1.
>
> * Extend Bayesian method to multi-layer transformer: The Bayesian method in [Zhang23] and our work assume perfect pre-training. We must study perfect pre-training to match real multi-layer transformers. This step is challenging.
>
> * Obtain generalization bound of ICL: This step is feasible following our framework in Appendix D.5. We need to derive the ϵ-stability of the multi-layer transformer.
>
> [Zhang23] What and how does in-context learning learn? bayesian model averaging, parameterization, and generalization
>
> [Hoover23] Energy Transformer
>
> > **C3**: Notation Inconsistencies
>
> **Response**
>
> We thank the reviewer for pointing out the potential notational inconsistency. $C_t$ and $R_t$ are written as matrices for clarity and convenience. In practice, they represent token sequences processed by the LLM in an autoregressive manner.
>
> > **C4**: Pseudo-Inverse Usage
>
> **Response**
>
> Thank you for raising this point. We acknowledge that the use of pseudo-inverses can indeed present numerical stability challenges. In this section, we primarily focus on the theoretical analysis of the in-context learning mechanism. We recognize that practical applications may require additional considerations, such as the implementation of regularization techniques, to address these stability concerns effectively. We add this to the second point of the limitation part (Appendix A.2).
>
> > **C5**: Generalization Bound
>
> **Response**
>
> * For the $\epsilon$-stability part, we want to clarify that under the reshaping framework, the softmax attention maintains its form and properties. The derivation of $\epsilon$-stability is not dependent on how the model performs memory reshaping. Hence, the attention mechanism satisfies $\epsilon$-stability.
>
> * The constants $(M, C_0, \Gamma)$ in our bound are standard parameters in theoretical analysis. $M$ represents the Lipschitz constant of the loss function. $\Gamma$ bounds the operator norm of weight matrices. $C_0$ is also a norm bound. These constants are commonly used in generalization bounds. They can be estimated empirically or bounded using standard techniques in learning theory.

---

> > ### Comment · Reviewer_HbbS · 2025-04-02
> >
> > I have carefully read all the reviewers' comments as well as the authors' responses, and my final opinion is to keep the score.

---

> > > ### Author Response · Authors · 2025-04-02
> > >
> > > Thank you for your careful consideration of both the reviewers' comments and the authors' responses. Really appreciate your thorough evaluation. Thanks very much!

---

### Official Review · Reviewer_uChw · 2025-03-12

**Overall Recommendation:** 4

**Summary:**

This paper considers, both theoretically and empirically, how a single attention head can be seen to perform in context learning (ICL) via the reshaping of an energy landscape shaped by the prompt. This work addresses the following topics:

1. Analyzing the effect of prompts through an interpretation of attention as the one-step Modern Hopfield Model (MHM) introduced by Ramsauer et al. 2020
2. Extending the work extends the work of [Zhang et al. 2023b] on Bayesian Model Averaging (BMA) for ICL to attention
3. Characterizing bounds on the performance of ICL based on the quantity and quality of prompt examples

**Claims And Evidence:**

C1. "Equivalent memory reshaping" of the pretrained weights in attention aptly describes ICL
C2. BMA explains how ICL works by performing BMA on the constructed hidden concept
C3. A one-layer attention model using reshaped memory has a clear generalization bound for ICL
C4. Equivalent memory reshaping in ICL explains model sensitivity to:
    (i) covariance shifts
    (ii) response accuracy
    (iii) similarity between prompts and test examples

Claims C1-C3 are fully supported by the strong theoretical contributions. I am not sufficiently confident in the empirical results to validate C4.

**Essential References Not Discussed:**

N/A

**Experimental Designs Or Analyses:**

The main experimental question is determining whether memory reshaping predicts observable behaviors of ICL, which the paper studies by considering a toy dataset consisting of a mixture of Gaussians. It is not clear from the presentation of the work how the experiments relate to the theory proposed earlier in the work, and it would be helpful if the theoretical claims (specifically, the bound) was verified in the experiments. See weaknesses.

**Methods And Evaluation Criteria:**

The method analyses is sound (though in its current form, limited to a single head of a single layer of attention). I have confusion about the experimental results. See weaknesses

**Other Comments Or Suggestions:**

- "Reshaped attention head" is mentioned for the first time in [L039 right col], but is not common knowledge and is not defined at this point in the paper.

### Typos
- MHM in the abstract is not defined, and MHM is not an appropriate acronym for "associative memory model" as mentioned in the introduction. It is first introduced in [L029 right col], but should be mentioned earlier.
- L368 "We use 4 different GM distribution" when only 3 are listed?

**Other Strengths And Weaknesses:**

### Strengths
- **Well organized**. The paper is well organized and written. Despite no familiarity in formally studying ICL, I was able to piece together the fundamental problem that the authors are trying to solve from their background sections.
- **Novel and meaningful contributions**. The main paper represents, as far as I know, a rigorous and novel perspective of ICL as memorization and energy minimization. Whether this scales to real Transformers with multiple heads of attention and multiple distinct layers is unclear (see weaknesses), but the approach is nonetheless meaningful.

### Weaknesses
- **Toy evaluations may not extend to language**. The theory and experiments consider ICL like a regression task, where the task of the model is only to predict the single next token. In traditional AR LLMs, ICL may be used to predict multiple tokens, and it is unclear how this paper generalizes to real scenarios and real-sized models.
- **Theory and evaluations are limited to a single layer, single attention head**. Again, the study simplifies Transformers dramatically. Unclear how it will generalize
- **Unclear takeaways from the experimental results**. It is not clear how to interpret the experimental results. Specifically, where do we see plotted the generalization bound against empirically observed behavior? What do the dashed gray lines mean in Figs. 1 & 2? Why does performance in Tables 1,2, & 3 peak at 45 ICL samples and decline (rather significantly) after? Please make it easier to understand how the experiments align with the theory


### Summary
There are core aspects of this work that I did not understand going into this paper, but the authors gave a proper background to the topic and presented their results in a clear and organized way. However, the paper has its flaws. The missing motivation for the experiments and the minimal discussion of the experimental results makes it unclear what the benefits of this approach are. It is also unclear if/how well the analyses will scale beyond the toy evaluations and models studied in this work.

Nonetheless, the strong theoretical analyses, the novel perspective, overal professionalism of the paper compel me to believe this paper is a proper fit for ICML. I am happy to increase my score if any of my misunderstandings can be clarified throughout the rebuttal process.

**Questions For Authors:**

Q1: Lemma 2.2: How does reshaping matrix $W$ play into the **energy function** of the MHM? Is it applied s.t. $\tilde{\xi}_k = W \xi_k$?

Q2: Sec 3.1: I'm a little confused about memory reshaping in the context of a prompt. Generally, autoregressive transformers take all prompt examples at once and the attention weight is not updated. Is this paper stating that attention processes all prompt tokens at once? Or that there is some kind of recurrence over the tokens?

Q3: How should I interpret $d_c$ and $d_r$ in [L064 right col]? Do you consider a case where a each query and response is only a single token (in which case $d_c = d_r = d$, the embedding size of the model)? This is not described.

Q4: Eq. (2.1) is a bit confusing. It seems to indicate that the LLM is pretrained only to predict queries $c_{t+1}$, but  autoregressive LLMs trained to predict all next tokens. Also, $C_t$ and $R_t$ are from the prompt, not necessarily the pretraining set. What is the connection here?

Q5: Can you explain why the "ghost examples" [L200-211] are necessary for testing this theory? Is it only to make it so the experiments vectorize across GPUs correctlY? I'm afraid I don't understand why ghost examples are necessary for the theory or the experiments.

Q6: It is not clear to me how the results in Table 1 use the GPT-2 model at all. Can you please explain the experiments in more detail?

**Relation To Broader Scientific Literature:**

The paper links Hopfield Networks and the in-context learning behavior of LLMs in an interesting way. As far as I know, this is a novel perspective that may produce quite interesting insight when scaled to larger models.

**Theoretical Claims:**

There are many theoretical claims and corresponding proofs in the paper. I did not have time to review many of them.

- Lemma 2.2: Memory reshaping in MHMs (by making the most similar stored pattern equal to the query) increases the separation ($\Delta$) between stored memories. This proof is correct.

---

> ### Author Rebuttal · Authors · 2025-04-01
>
> Thanks for your detailed review. We have revised our draft and addressed all concerns. **The revised version (changes marked in BLUE) and the code for external experiments are available in this [anonymous Dropbox folder](https://www.dropbox.com/scl/fo/9m982gnk45wc1w705wjsa/AKYGxykAcCFGADGQEHCN218?rlkey=h03t7ipyv8qbt9bvesfivxcsw&st=ze6gjwhu&dl=0).**
>
> > **C1**: Single next token
>
> **Response**
>
> We provide details on how to extend our method to multiple tokens in the 5th term of the limitation section (Appendix A.2) of the revised version. Please see the revised version for details.
>
> > **C2**: Simplifies Transformers dramatically
>
> **Response**
>
> We provide an overview of how to extend our analysis to multi-layer models. The detailed discussion appears in the 4th term of the limitation section (Appendix A.2) of the revised version. Please see the revised version for details.
>
> > **C3**: Interpretation of the experimental results
>
> **Response**
>
> * **Generalization Bound**: Figures 1, 2, and 3 show that the R-squared value increases when the number of in-context examples is below 50—the same number used during pretraining. This supports our observation that the generalization bound improves with more prompt examples.
>
> * **Dashed Gray Lines**: These indicate the threshold of 50 in-context examples, which matches the pretraining setup.
>
> * **Performance Decline Beyond 50**: During pretraining, we use a prompt length of 50 (Appendix E.1). When evaluated on longer contexts, GPT-2 shows a drop in R-squared. We conjecture that this is due to its use of absolute positional encodings (Remark 4.1).
>
> > **C4**: Motivation and benefits of the experiments
>
> **Response**
>
> **Motivation for the experiments**:
> We design these experiments to validate three key properties of ICL: the model’s sensitivity to (i) covariance shifts, (ii) response accuracy, and (iii) the similarity between prompts and test examples. We then provide theoretical explanations for these properties based on our results. Our empirical and theoretical validations provide the practical guidelines for improving ICL performance.
>
> **Benefits of this approach**:
> Memory reshaping provides a theoretical explanation for three key properties of ICL.
>
> **Our results can scale beyond the toy evaluations well**. In the revised version, we extend our experimental evaluation to decision trees, 2-layer neural network tasks, and practical sentiment classification tasks. Please refer to Section 4.1.4 for details.
>
> > **C5**: Typos
>
> **Response**
>
> We have added the definition and corrected the typos in the revised version. The reshaped attention head refers to a linear transformation of the key and value matrices.
>
> > **Q1**:
>
> **Response**
>
> Yes, we apply it as $\tilde{\xi}_k = W \xi_k$.
>
> > **Q2**:
>
> **Response**
>
> We consider that transformers take all prompt examples at once, and the attention weights are not updated.
>
> > **Q3**:
>
> **Response**
>
> Yes, we consider the single token case.
>
> However, there is no constraint requiring $d_c = d_r = d$, as stated in the second paragraph of Section 3.1.
>
> Our method can also be extended to multi-token settings, with details provided in the Limitations section (Appendix A.2) of the revised version.
>
> > **Q4**:
>
> **Response**
>
> Yes, in Eq. (2.1), we do not consider predicting multiple tokens.
>
> In Eq. (2.1), we denote the pretrained LLM as $P_{\theta}$, where $\theta$ represents the pretrained model parameters. $C_t$ and $R_t$ are independent of $\theta$, Eq. (2.1) simply describes using $P_{\theta}$ to predict $c_{t+1}$ based on the prompt $C_t$ and $R_t$.
>
> > **Q5**:
>
> **Response**
>
> We apologize for the confusion. The answer is No.
>
> The importance of the ghost example is for mathematical consistency. Here are some clarifications.
>
> * As shown in Remark 3.1, we use the proxy attention based on the BMA formulation, and this is not the actual attention in LLM.
>
> * Consider the setting where we only have the input $c_{t+1}$ without $C_t$ and $R_t$. For the proxy attention mechanism equation (3.1), $K_t$ and $V_t$ have no meaning.
>
> * Therefore, we construct $t$ ghost examples and prepend them before $c_{t+1}$.
>
> * The aims are that the Eq. (3.1) is reasonable and the t ghost examples do not change the response of the $c_{t+1}$ in ICL.
>
> > **Q6**:
>
> **Response**
>
> We apologize for the confusion. Due to space constraints, we provide the full details in Appendix E. Below, we summarize the core idea:
>
> * Pretraining of GPT-2: We use the GPT-2 architecture but pretrain it on synthetic data generated from a linear regression setting. Each pretraining sample contains 50 query-response pairs. We apply MSE loss as training loss to predict the response for each pair.
>
> * As described in Appendix E.2, we evaluate model performance when the number of in-context examples is less than, equal to, or greater than 50. We use R-squared as the evaluation metric and report values at 15, 30, 45, 60, and 75 examples in Table 1. The full R-squared curve is shown in Figure 3.1(a).

---

> > ### Comment · Reviewer_uChw · 2025-04-03
> >
> > Thank you for answering my questions and updating your draft for clarity. I believe the novel contributions and perspective of this work outweigh the limitations (which are now clearly discussed in Appendix A.2 of the updated draft). I am satisfied with the updated changes in the manuscript and believe the paper is now a stronger submission. I am updating my score to a 4.

---

> > > ### Author Response · Authors · 2025-04-03
> > >
> > > Thank you for your thorough review and for the improvement in the score. We are pleased to have addressed your concerns and truly appreciate your feedback. Thanks very much!

---

### Official Review · Reviewer_oD5q · 2025-03-14

**Overall Recommendation:** 3

**Summary:**

**Main Findings**:

1. Memory Reshaping: This paper proposes that ICL can be understood as a process of memory reshaping within the Hopfield model framework. Specifically, the input prompt examples can reshape the energy landscape of the probabilistic energy-based memory model, thus relocating the distribution of local minima.

2. Bayesian Model Averaging with Reshaped Memory: The authors formulate the explicit expression, demonstrating that ICL performs conditional estimation based on this hidden concept. Besides, softmax attention can approximate Bayesian Model Averaging.

3. Generalization Bound: This paper derives a generalization bound for ICL in a one-layer attention model, demonstrating that the performance of ICL improves with an increasing number of prompt examples.

**Main Algorithmic/Conceptual Ideas**:

1. Memory Reshaping Mechanism: This paper introduces the concept of memory reshaping, where the pre-trained attention mechanism's weights are linearly transformed by the prompt examples.

2. Hidden Concept Learning: The authors argue that ICL implicitly learns a hidden concept from the prompt examples, which is then used for conditional estimation. This hidden concept is shown to be a shared property among the prompt examples.

3. Hopfield Model Interpretation: This paper provides a new perspective on how transformers process and adapt to context by framing ICL as memory retrieval in the modern Hopfield model.

**Main Results**:

1. Theoretical Results: This paper provides mathematical formulations and proofs for memory reshaping, the connection to BMA, and the generalization bound.

2. Empirical Results: The experiments confirm the theoretical predictions, showing clear trends in ICL performance with respect to covariance shifts, response accuracy, and prompt-test query similarity.

**Claims And Evidence:**

The claims made in the paper are generally well-supported by clear and convincing evidence. The theoretical analysis provides a solid foundation, and the empirical experiments validate the key findings.

**Essential References Not Discussed:**

No missing related works.

**Experimental Designs Or Analyses:**

The experimental designs and analyses in the paper are generally sound and well-structured, providing clear evidence for the theoretical claims. The choice of tasks, models, and metrics is appropriate for the problem at hand. However, there are areas where further exploration could strengthen the findings and enhance the generalizability of the results.

1. Task Diversity: Testing the framework on more complex tasks (e.g., classification, natural language inference) could provide deeper insights into the behavior of ICL.
2. Model Diversity: Evaluating the framework on more advanced models (e.g., GPT-3, GPT-4) could strengthen the generalizability of the findings.
3. Real-World Data: Using real-world datasets and more diverse distribution shifts could provide a more realistic evaluation of ICL.

**Methods And Evaluation Criteria:**

The proposed methods and evaluation criteria in the paper are highly suitable for the problem of understanding ICL in LLMs. The theoretical framework is rigorous, and the experimental design effectively validates the claims. The choice of metrics and baselines provides a clear and interpretable evaluation framework.

However there is still room for further exploration in more diverse tasks and more models. Considering that the model used in the paper  (GPT-2) and the task (linear regression) are rather simple, whether the findings are general to more complex models and tasks remains to be proved.

**Other Comments Or Suggestions:**

No further comments.

**Other Strengths And Weaknesses:**

**Strength**:

1. Novel Interpretation of ICL: The paper offers a new perspective on In-Context Learning (ICL) by interpreting it as memory retrieval in the modern Hopfield model. This interpretation is creative and provides a fresh understanding of how ICL works.

2. Theoretical Contributions: The paper makes significant theoretical contributions by deriving a generalization bound for ICL and showing how softmax attention can approximate Bayesian Model Averaging. These results provide a deeper understanding of the underlying mechanisms of ICL.

3. Clarity: The paper is well-organized and clearly presented. The theoretical framework is explained in detail, and the mathematical derivations are rigorous and easy to follow.

**Weaknesses** :

1. Limited Scope: The theoretical analysis focuses on a single-layer attention model, which may not fully capture the complexity of modern LLMs with multiple layers. Extending the analysis to multi-layer models would strengthen the findings.

2. Linear Regression Task: The experiments are limited to a linear regression task, which is relatively simple. Testing the framework on more complex tasks would provide a more comprehensive evaluation of ICL.

3. Lacking novel guidelines: This paper provides a novel understanding of ICL and get several guidelines. However, most of the guidelines are rather intuitive, like "As demonstrated in Section 4.1.3, selecting prompts similar to the test query enhances in-context learning performance." and "By Theorem 3.2, increasing the number of relevant prompt examples reduces the ICL generalization error". These guidelines of course can prove the proposed understanding, but may lack contributions. If the authors can further reach a novel guideline based on the proposed interpretation theoretically, and prove it empirically, this paper will bring more contributions to the research area.

**Questions For Authors:**

1. Is the proposed interpretation general to more complicated models and more challenging tasks?
2. Are there more further guidelines your interpretation can provide?

**Relation To Broader Scientific Literature:**

This paper offers a new perspective on In-Context Learning by interpreting it as memory retrieval in the modern Hopfield model. This novel interpretation can bring contributions to this area.

However, the guidelines provided can be further improved. Most of the guidelines are quite intuitive like 'select similar examples' and 'use more similar examples'. This resricts the contributions to the broader scientific literature. If the authors could provide more novel guidelines, I believe this will bring more contributions to not only the interpretability of ICL and LLM, but can also help the users of LLM.

**Theoretical Claims:**

The theoretical claims made in the paper are supported by clear and convincing evidence. The proofs are mathematically rigorous and logically sound, and the empirical results validate the theoretical predictions.

---

> ### Author Rebuttal · Authors · 2025-04-01
>
> Thanks for your detailed review. We have revised our draft and addressed all concerns. **The revised version (changes marked in BLUE) and the code for external experiments are available in this [anonymous Dropbox folder](https://www.dropbox.com/scl/fo/9m982gnk45wc1w705wjsa/AKYGxykAcCFGADGQEHCN218?rlkey=h03t7ipyv8qbt9bvesfivxcsw&st=ze6gjwhu&dl=0).**
>
> > **C1**: Limited Scope
>
> **Response**
>
> Thanks for your insightful comment. We provide an overview of how to extend our analysis to multi-layer models. The detailed discussion appears in the Section A.2 of the revised version, and we restate it here.
>
> We regard the multi-layer transformer as iteratively retrieving memory patterns in the modern Hopfield model (MHM) through the following four steps:
>
> * Design energy function for multi-layer MHM: We design an energy function matching the multi-layer transformer [Hoover23]. The memory pattern needs retrieval after reshaping and iteration. This step is challenging.
>
> * Give clear expression of memory reshaping on model weights: This step is feasible following our framework in Sec. 3.1.
>
> * Extend Bayesian method to multi-layer transformer: The Bayesian method in [Zhang23] and our work assume perfect pre-training. We must study perfect pre-training to match real multi-layer transformers. This step is challenging.
>
> * Obtain generalization bound of ICL: This step is feasible following our framework in Appendix D.5. We need to derive the ϵ-stability of the multi-layer transformer.
>
> [Zhang23] What and how does in-context learning learn? bayesian model averaging, parameterization, and generalization
>
> [Hoover23] Energy Transformer
>
> > **C2 & Q1**: Linear Regression Task: Is the proposed interpretation general to more complicated models and more challenging tasks?
>
> **Response**
>
> Thanks for your valuable question. The answer is **Yes**.
>
> In the revised version, we extend our experimental evaluation to cover more complex tasks. Specifically, we include new results using GPT-2 on decision tree and 2-layer neural network tasks, following a similar setup as in the linear regression experiments. Additionally, we incorporate experiments with the GPT-J model on the real-world “TweetEval: Hate Speech Detection” dataset to evaluate performance on a practical sentiment classification task. Please refer to Section 4.1.4 for details.
>
> > **C3 & Q2**: Lacking novel guidelines: Are there more further guidelines your interpretation can provide?
>
> **Response**
>
> Thank you for your comment. Our interpretation indeed offers further guidelines that help explain phenomena like hallucination in LLMs and avoid hallucination. For example:
>
> * **Biased Hidden Concept**:
> Eqs. (2.2)–(2.4) illustrate how the hidden concept influences the in-context learning response under a Bayesian approach. In Lemma 3.1, we derive an explicit expression for this hidden concept from the memory reshaping perspective. Irrelevant or inconsistent prompt examples create a biased hidden concept, which in turn leads to hallucination.
>
> * **Insufficient Separation of Memory Patterns**:
> The modern Hopfield model (MHM) shows that the separation of memory patterns affects retrieval error. We conjecture that unreasonable prompt examples reduce the separation between different memory patterns, increasing the retrieval error and contributing to hallucination.
>
> Our framework provides a clear and practical guideline: to reduce hallucination, prompt examples should be relevant and consistent with the test input. To validate this, we refer to the experiments on the model’s sensitivity to the similarity between prompts and test examples. These experiments can be used here, including those with the GPT-2 model on linear regression, decision tree, and 2-layer neural network tasks.

---

> > ### Comment · Reviewer_oD5q · 2025-04-04
> >
> > The rebuttal solves most of my concerns. I keep my rating of weak accept.

---

> > > ### Author Response · Authors · 2025-04-05
> > >
> > > Thank you for your careful consideration of the responses. Really appreciate your thorough evaluation. Thanks very much!

---

### Official Review · Reviewer_yDv9 · 2025-03-19

**Overall Recommendation:** 2

**Summary:**

This paper proposes a model to interpret in-context learning as associative retrieval. The paper argues that in-context learning can be seen as a linear transformation of key-value weights. Under a GLM assumption, their modified attention construction (BMA Attention) is shown to approximate softmax attention and converge to the presumed target, in the limit of infinite in-context examples in the prompt. Their theory is supported using a linear regression experimental setup within the ICL setting.

**Claims And Evidence:**

There are a lot of claims made in the paper, but often, the corresponding analysis/results either does not present strong enough evidence to establish them or connect them back well to the larger picture. I list these below:

- Connection to Associative Memory: In line 35, col 2, it is claimed that the paper defines an energy function for LLMs to interpret it as a Hopfield model and interpret ICL as reshaping attention heads. However, in the following discussion in 3.1, I see no mention of an energy function for the LLM, but only an analysis using a modified attention mechanism. Thus, the only connection to retrieval seems to be the proposed memory reshaping mechanism, which appears as a tenuous link to reducing retrieval error in Lemma 2.2.

- Relevance of Hidden Concepts: I am unsure as to how the notion of hidden concepts introduced in section 2.2 fits the larger context. Specifically, the generative model (with a GLM assumption) is used in section 3.2 to yield a certain prediction model that approximates z* via bar{z}t. However, I do not follow why the true target v{t+1} would be a function of the synthetic prompt form in 3.7, or what intuition does 3.9 offer within the larger context of the associative retrieval.

- Mathematical Impreciseness: In 3.4, it is implicitly assumed that there exists a linear transformation to go from the ghost prompts to the true keys and values, which will simply not be true if the ghost examples constitute of lower rank matrices. The discussion should note that the pseudo-inverse structure only minimizes the approximation error to the true matrix and does not directly approximate it. Similarly, in Assumption 3.1, (k, v) are assumed to be iid, without defining any underlying sampling distribution, either over them or the original (c, r) pairs. In addition, I am not even sure if this assumption holds true either in their model (which is a 1-layer transformer operating directly on embeddings) because of the weight matrix or the joint optimization process over the ghost examples that naturally introduces dependencies or in practice, since the justification that layer norm will somehow achieve this is not clear in the sense that layer norm only operates on a per token basis and affects no intra batch dependencies.

- Missing Proof Sketches: The proof sketch is missing for certain theorems and lemmas. But more importantly, the intuition on how to interpret this result in the larger context of the paper is often missing as well, meaning that as a reader, it is hard to get a high-level understanding of what's being said.

It is certainly possible that I missed some links that the authors had in mind while writing the paper. However, I still stress that they should be made more explicit to make the paper more coherent for the general reader.

**Essential References Not Discussed:**

There is some missing discussion of related literature and what exactly are the paper's unique contributions.

- The paper borrows the setup heavily from [1], but the line to demarcate their own novel results from the prior work is not made very clear for the reader.
- Discussion of ML theory work [see 2, 3 as a starting point] that interprets Transformers as associative memories, and how it differs from their work is missing.


References:

[1] What and How does In-Context Learning Learn? Bayesian Model Averaging, Parameterization, and Generalization - Zhang et al., 2023

[2] Birth of a Transformer: A Memory Viewpoint - Bietti et al., 2023

[3] Scaling Laws for Associative Memories - Cabannes et al, 2024

**Experimental Designs Or Analyses:**

See Methods and Evaluation Criteria

**Methods And Evaluation Criteria:**

While the experimental validation of the proposed theory is not the main contribution of the paper, there are some issues with it as well.

- The paper attempts to validate the theoretical claims with linear regression ICL experiments, interpreting their findings from a memory shaping viewpoint. However, the experimental setup is rather general, and the resulting trends would be predictable even without any intuition of reshaping. I understand that a claim like this is not easy to establish in general, but the current setup does not lend a lot of extra credence either.

- The setting of linear regression is perhaps not the most informative for LLMs, and the authors should consider other experiments on language centric datasets [see setups in 1, 2] for a stronger justification. Additionally, certain details for the experimental setup (such as how to measure similarity between examples in 4.1.3) are missing and need to be specified.

References:

[1]:  In-Context Learning with Many Demonstration Examples - Li et al., 2023

[2]: Everything Everywhere All at Once: LLMs can In-Context Learn Multiple Tasks in Superposition - Xiong et al., 2024

**Other Comments Or Suggestions:**

N/A

**Other Strengths And Weaknesses:**

N/A

**Questions For Authors:**

N/A

**Relation To Broader Scientific Literature:**

The paper belongs to the line of work which interprets certain mechanisms in transformers as a retrieval in associative memories, with the primary contribution being an explicit mechanism of memory reshaping to explain ICL.

**Theoretical Claims:**

Proofs were checked on a high level, with no obvious issues found.

---

> ### Author Rebuttal · Authors · 2025-04-01
>
> Thanks for your detailed review. We have revised our draft and addressed all concerns. **The revised version (changes marked in BLUE) and the code for external experiments are available in this [anonymous Dropbox folder](https://www.dropbox.com/scl/fo/9m982gnk45wc1w705wjsa/AKYGxykAcCFGADGQEHCN218?rlkey=h03t7ipyv8qbt9bvesfivxcsw&st=ze6gjwhu&dl=0).**
>
> > **C1**: No mention of an energy function
>
> **Response**
>
> Thanks for the comment.
>
> We use a toy model (one-layer attention) to study the in-context learning of LLM. For the one-layer attention, the energy function of this toy model is equivalent to that of MHM. We argue such an atomic setting is prevalent for analytical feasibility in literature [XIe22, Zhang23].
>
> [Xie22] An Explanation of In-context Learning as Implicit Bayesian Inference
>
> [Zhang23] What and How does In-Context Learning Learn? Bayesian Model Averaging, Parameterization, and Generalization
>
> > **C2**: Notion of hidden concepts
>
> **Response**
>
> We apologize for the confusion.
>
> The main point arises from memory reshaping in Eq. (3.5). To interpret ICL using memory reshaping, we must incorporate synthetic prompts, as noted in Remark 3.2. In Eq. (3.7), we aim to demonstrate the formulation of Eq. (3.6) with memory reshaping, thus including the synthetic prompts.
>
> Eq. (3.9) explicitly formulates Lemma 2.1, relating to the mathematical formulation of ICL. We incorporate memory reshaping in Eq. (3.9) to demonstrate how it influences the hidden concept and subsequently impacts ICL performance.
>
> > **C3**: Linear transformation and pseudo-inverse
>
> **Response**
>
> Thanks for your insightful comments. **We acknowledge these limitations and have added them to the 2nd and 3rd terms of limitation part (Appendix A.2 of the revised version).** Please refer to the revised version for details. We promise to add a technical remark to the main text in the final version if space allows.
>
> Furthermore, it is acceptable that Assumption 3.1 does not define an explicit underlying sampling distribution, as the theoretical analysis remains valid under general distributional conditions without requiring a specific form.
>
> > **C4**: Missing proof sketches
>
> **Response**
>
> We have added proof sketches for Lemma 2.1, Lemma 2.2, Lemma 3.1, and Theorem 3.1 ahead of the official proofs in the Appendix. We also provide the following high-level overview of our theoretical results.  We promise to move these to the main text in the final version, if space allows.
>
> **High-Level Understanding of Theoretical Results:**
>
> * Lemma 2.1: Interprets In-Context Learning (ICL) in Large Language Models (LLMs) through Bayesian Model Averaging (BMA).
>
> * Lemma 2.2: Shows that memory reshaping in Modified Hopfield Models (MHMs) reduces retrieval errors. Given the equivalence of the energy functions between MHMs and LLMs, this motivates us to apply memory reshaping to interpret ICL in LLMs.
>
> * Lemma 3.1: Integrates memory reshaping into the Bayesian Model Averaging framework, providing a more explicit interpretation of ICL under the Gaussian linear case than Lemma 2.1.
>
> * Theorem 3.1: Establishes the connection between Bayesian Model Averaging and Softmax Attention, showing that the explicit interpretation provided by Lemma 3.1 corresponds exactly to ICL within the attention mechanism.
>
> > **C5**: The experimental setup is rather general
>
> **Response**
>
> Thank you for your helpful comment. We agree this is a limitation and have added it to the **6-th item of limitations section (Appendix A.2 of the revised version).** Please refer to the revised version for details. We also suggest one possible solution. However, verifying this solution is difficult. We leave that for future work.
>
> > **C6**: Linear regression is simple
>
> **Response**
>
> Thanks for your valuable comment.
>
> In the revised version, we extend our experimental evaluation to cover more complex tasks. Specifically, we include new results using GPT-2 on decision tree and 2-layer neural network tasks, following a similar setup as in the linear regression experiments. Additionally, we incorporate experiments with the GPT-J model on the real-world “TweetEval: Hate Speech Detection” dataset to evaluate performance on a practical sentiment classification task. Please refer to Section 4.1.4 for details. The results follow a similar pattern to the linear regression setting, consistent with our theoretical predictions.
>
> We apologize for the confusion. We use the cosine similarity between different queries as the similarity measure. We have added this and other details to the revised version.
>
> > **C7**: Essential references
>
> **Response**
>
> Thank you for your question. We have included a comparison with these three works in the revised version. Please see Appendix A.1 of the revised version for details.

---

> > ### Comment · Reviewer_yDv9 · 2025-04-07
> >
> > I thank the authors for their rebuttal; I have taken a look at the updated submission and appreciate the efforts to address my concerns. I still have a few lingering issues about the presentation:
> >
> > C1) I am aware of the connection between MHMs and one layer attention models. However, there is no correspondence demonstrated between the defined equation for retrieval (eqn. 3.8/3.9) and the energy function of MHMs (or any energy function for that matter). Therefore, I again stress that in the context of this submission, claiming that "we define an energy function for LLMs" is misleading for the reader. The authors should either establish this or consider rephrasing this.
> >
> > C3) With regards to keys and values iid assumption, I do not mean to say that the authors should specify a particular distribution. Rather, since K, V are functions of the prompts and query tokens, the authors should state what is being assumed over the actual sampling distribution. I would also urge the authors to at least include the theoretical limitations as remarks next to the pertinent results, as opposed to deferring it to the appendix for providing the readers with appropriate context.
> >
> > With these changes, I'd be happy to improve my recommendation for the paper.

---

> > > ### Author Response · Authors · 2025-04-07
> > >
> > > Thank you for your detailed review and feedback.  We appreciate your willingness to consider improving your recommendation.
> > >
> > > We have revised our draft and addressed your concerns. The revised version (changes marked in BLUE) and the code for external experiments are available in this **[anonymous Dropbox folder](https://www.dropbox.com/scl/fo/9m982gnk45wc1w705wjsa/AKYGxykAcCFGADGQEHCN218?dl=0&e=2&rlkey=h03t7ipyv8qbt9bvesfivxcsw&st=ze6gjwhu)**.
> > >
> > > ---
> > >
> > > >**C1**: Revision of the Energy Function Description.
> > >
> > > **Response**
> > >
> > > Thank you for your suggestion regarding the description of the energy function. We agree with your assessment and have made the following revisions to the manuscript to ensure clarity:
> > >
> > > * `line 032-034, column 2`: Revised "we define an energy function for LLMs" to "we parametrize this energy function";
> > > * `line 129-131, column 2`: Revised "The energy function of LLM" to "The energy function of one-layer attention".
> > >
> > > >**C2**: Clarification on Assumptions and Theoretical Limitations.
> > >
> > > **Response**
> > >
> > > Thank you for your clarification on the iid assumption related to keys and values. We apologize for any previous confusion and have provided additional details in the manuscript as follows:
> > >
> > > * **Considering the actual sampling distribution, Assumption 3.1 further assumes that $\\{c_t, r_t\\}^T_{t=1}$ are independently and identically distributed.**
> > >
> > > * We have added this and theoretical limitations to Remark 3.3 (`line 178-183, column 2`) and Remark 3.6 (`line 268-274, column 1; line 220-222, column 2`).
> > >
> > > ---
> > >
> > > We hope these revisions address your concerns and contribute to the manuscript's improvement. We are grateful for your insightful feedback.

---

### Decision · Program_Chairs · 2025-05-01

**Decision:**

Accept (poster)

**Comment:**

This is borderline paper. It formalizes in-context learning as a memory retrieval in a modern Hopfield network. After the discussion with the reviewers I believe that the reasons to accept this work outweigh reasons to reject. The camera-ready version should use clear language referring to the energy function for a one step attention operation rather than the entire LLM.